# Closing the Plio-Pleistocene $^{13}$C cycle in the 405-kyr periodicity by isotopic signatures of geological sources

Peter Köhler[1]

[1]Alfred-Wegener-Institut Helmholtz-Zentrum für Polar- und Meeresforschung, P.O. Box 12 01 61, 27515 Bremerhaven, Germany

**Correspondence:** Peter Köhler (peter.koehler@awi.de)

**Version: March 20, 2025**

**Abstract.**

The $^{13}$C cycle of the Plio-Pleistocene, as recorded in $\delta^{13}$C of benthic foraminifera, has power in periodicities related to the long eccentricity cycle of 405-kyr that is missing in corresponding climate records (e.g. $\delta^{18}$O). Using a global carbon cycle model I show in an inverse approach that the long eccentricity in $\delta^{13}$C might have been caused by variations in the isotopic signature of geological sources, namely of the weathered carbonate rock ($\delta^{13}$C$_{\mathrm{rock}}$) or of volcanically released CO$_2$ ($\delta^{13}$C$_{\mathrm{v}}$). This closure of the $^{13}$C cycle in these peridicities also explains the offset in atmospheric $\delta^{13}$CO$_2$ seen between the penultimate and the last glacial maximum. The necessary isotopic signatures in $\delta^{13}$C$_{\mathrm{rock}}$ or $\delta^{13}$C$_{\mathrm{v}}$ which align my simulations with reconstructions of the $^{13}$C cycle on orbital timescales have most power in the obliquity band (41-kyr) suggesting that land ice dynamics are the ultimate cause for these suggested variations. Since the Asian monsoon as reconstructed from speleothems has also an obliquity-related component and since precipitation or runoff is one main driver for local weathering rates it is possible that these proposed changes in weathering are indeed, at least partly, connected to the monsoon as previously suggested. Alternatively, the suggested impact of land ice or sea level on volcanic activity might also be influential for the $^{13}$C cycle. This indirect influence of ice sheets on the long eccentricity cycle in $\delta^{13}$C implies that these processes might not have been responsible for the 405-kyr periodicity found in times of the pre-Pliocene parts of the Cenozoic that have been largely ice free in the northern hemisphere.

# 1 Introduction

The long eccentricity cycle with a periodicity of 405-kyr is for the last 5 Myr missing in most climate signals, e.g. in the benthic $\delta^{18}O$ (Lisiecki and Raymo, 2005), but is imprinted on the global carbon cycle since significant power in it is contained in benthic $\delta^{13}C$ (e.g. Mix et al., 1995b; Wang et al., 2010). For earlier parts of the Cenozoic the 405-kyr cycle is not only found in $\delta^{13}C$ but also in $\delta^{18}O$ (Pälike et al., 2006; Zeebe et al., 2017; De Vleeschouwer et al., 2020; Westerhold et al., 2020). This difference between climate and carbon cycle makes the interpretation of the Plio-Pleistocene $^{13}C$ cycle and related interpretations with respect to processes responsible for changes in atmospheric $CO_2$ challenging.

Some interpretations of the impact of long eccentricity on $\delta^{13}C$ have been put forward. Russon et al. (2010) used a carbon cycle box model to analyse how oceanic processes, e.g. small changes in nutrient availability or in the ratio of organic matter to $CaCO_3$ in the export production, might have caused these $\delta^{13}C$ changes. They already point out that changes in only the $^{13}C$ cycle but not atmospheric $CO_2$, and therefore global climate, are difficult to explain. Ma et al. (2011) used a slightly different box model and proposed that eccentricity impacts on the weathering intensity and nutrient supply ultimately changed $\delta^{13}C$ in $\sim$400-kyr periodicity during the Miocene. Paillard (2017) used a conceptual model and suggested that riverine organic carbon inputs into the ocean caused by sea level related erosion at river mouths might be responsible for $\sim$400-kyr variability in $\delta^{13}C$ of the last 4 Ma. In a data review Wang et al. (2014) also proposed that the origin of this low $\delta^{13}C$ frequency is related to riverine input, but suggested that the underlying process is the monsoon intensity.

While the long-term changes in $\delta^{13}C$ in the ocean have long been known (e.g. Mix et al., 1995b) the record of $\delta^{13}CO_2$ from ice cores is still relatively new and with 155 kyr rather short (Eggleston et al., 2016). Upcoming new ice core data across Termination IV (around 340 kyr BP) offer another $\sim$10 kyr long snapshot of changes in $\delta^{13}CO_2$ (Krauss et al., 2025). When $\delta^{13}CO_2$ data across Termination II were first measured (Schneider et al., 2013) an apparent offset between the Penultimate and the Last Glacial Maximum (PGM and LGM, respectively) of +0.45‰ prevented a straight-forward interpretation of the data. Among other possibilities the authors speculated that geological processes, namely changes in the isotopic ratio of carbonate weathering or in the contributions from volcanic input fluxes, might explain this offset in $\delta^{13}CO_2$. However, it was not yet discussed in the ice core community (Schneider et al., 2013; Eggleston et al., 2016) if the PGM-to-LGM offset in $\delta^{13}CO_2$ and the long eccentricity cycle in benthic $\delta^{13}C$ are related to each other.

Here, I use the well-established global carbon cycle box model BICYCLE-SE (Köhler and Munhoven, 2020) to test in detail, if these geological processes proposed by Schneider et al. (2013) might indeed be responsible for PGM-to-LGM offset in $\delta^{13}CO_2$ and eventually also for the changes in the $^{13}C$ cycle related to the long eccentricity. For that effort I first reanalyse previously published simulations which have been performed with an updated $^{13}C$ cycle over the time window covered by $\delta^{13}CO_2$ (Köhler and Mulitza, 2024). In that study atmospheric $\delta^{13}CO_2$ was in some simulations already prescribed by the ice core data, and not internally calculated in the model. This approach indirectly introduced some long-term variability in $\delta^{13}C$ to the whole $^{13}C$ cycle, while the boundary conditions changing the main carbon cycle and $CO_2$ were not modified. I first analyse if and how these changes in $^{13}C$ necessary for $\delta^{13}CO_2$ to agree with ice cores reconstructions might have been caused by the two geological processes of interest. I then add further scenarios to my assessment, in which one variable of

the $^{13}$C cycle in the model is not prescribed by data, but only weakly nudged to a $\delta^{13}$C time series. This nudging approach reduces the amount of necessary adjustments considerably and is first applied to atmospheric $\delta^{13}CO_2$, and then expanded to deep ocean $\delta^{13}$C in order to be able to cover simulations longer than 155 kyr. Building on earlier work I then extend my investigation to simulations covering the ice core time window of the last 800 kyr (Köhler and Munhoven, 2020) and finally the last 5 Myr covering most of the Plio-Pleistocene (Köhler, 2023). Such an approach of nudging the model to data is an inverse investigation, which offers new insights, not directly (or only under a time consuming effort) available from classical forward modelling. While in the latter well-defined hypothesis are tested by changing the boundary conditions or processes beforehand in simulated scenarios the inverse approach offers solutions one might not have been looking for (e.g. unexpected frequencies in the variables of investigation). As a side effect my nudging approach might also increase the simulated glacial-interglacial (G-IG) amplitudes in $\delta^{13}$C in the earlier parts of the Plio-Pleistocene and bring them in better agreement with reconstructions (Köhler and Bintanja, 2008).

## 2 Methods

### 2.1 Data

To perform this analysis the following data (reconstruction) are needed: Plio-Pleistocene deep ocean time series of benthic $\delta^{13}$C and $\delta^{18}$O, all available atmospheric $\delta^{13}CO_2$ from ice cores and late Pleistocene surface ocean (planktic) $\delta^{13}$C, and estimates on the variability in the $\delta^{13}$C signature from carbonate rock weathering and volcanic $CO_2$ outgassing. These data are described in detail in this subsection.

#### 2.1.1 Plio-Pleistocene deep ocean time series

Due to the large volume of the Pacific ocean whole ocean changes in $\delta^{13}$C are to a first order approximated with benthic $\delta^{13}$C data from the deep Pacific. Under this premise a deep Pacific $\delta^{13}$C stack consisting of six sediment cores (Lisiecki, 2014) illustrates the variability in oceanic $\delta^{13}$C over the last 3 Myr (Fig. 1b), which is in its long-term variability, but with higher short-term scatter and a by 0.1‰ lower long-term mean, nicely matched by $\delta^{13}$C from ODP846 (Mix et al., 1995a; Shackleton et al., 1995), which is also one of the cores contributing to the stack. This core ODP846 (3307 m water depth, 91°W close to the equator) extends further back in time and is used here as published in Poore et al. (2006) in order to cover a 5 Myr long time window (Fig. 1b).

Stacking of cores always introduces a smoothing effect. Therefore, the width of the data distribution of the both time series converge to each other ($\sigma \approx 0.20$), if a 5-points-running mean is applied to the ODP846 data (Fig. 1c), which in its raw data is more widely distributed ($\sigma = 0.26$). Both deep Pacific $\delta^{13}$C time series contain significant power in periods longer than 100 kyr in their wavelets (Fig. 1d,e), not only around 405-kyr ($\sim$500-kyr in the last 1 Myr), but also in-between the eccentricity bands ($\sim$200-kyr periodicities). While the latter variability is not known from orbital forcing, it has already been detected before in terrestrial sediments of the early Paleocene, and might be a sub-harmonic of the 405-kyr periodicity (Hilgen et al.,

2020). Alternatively, this power might be related to the 173-kyr periodicity, an amplitude modulation of the 41-kyr of obliquity, which is also found in the Miocene-Pliocene of Asian monsoon (Zhang et al., 2022) and in total organic carbon during the Mesozoic and Cenozoic (Huang et al., 2021). Power in periodicity in-between 100 and 405 kyr during the last 5 Myr are also contained in previous wavelet analysis of so-called benthic $\delta^{13}$C mega-splices, but they have never been discussed in more

detail (De Vleeschouwer et al., 2020; Westerhold et al., 2020).

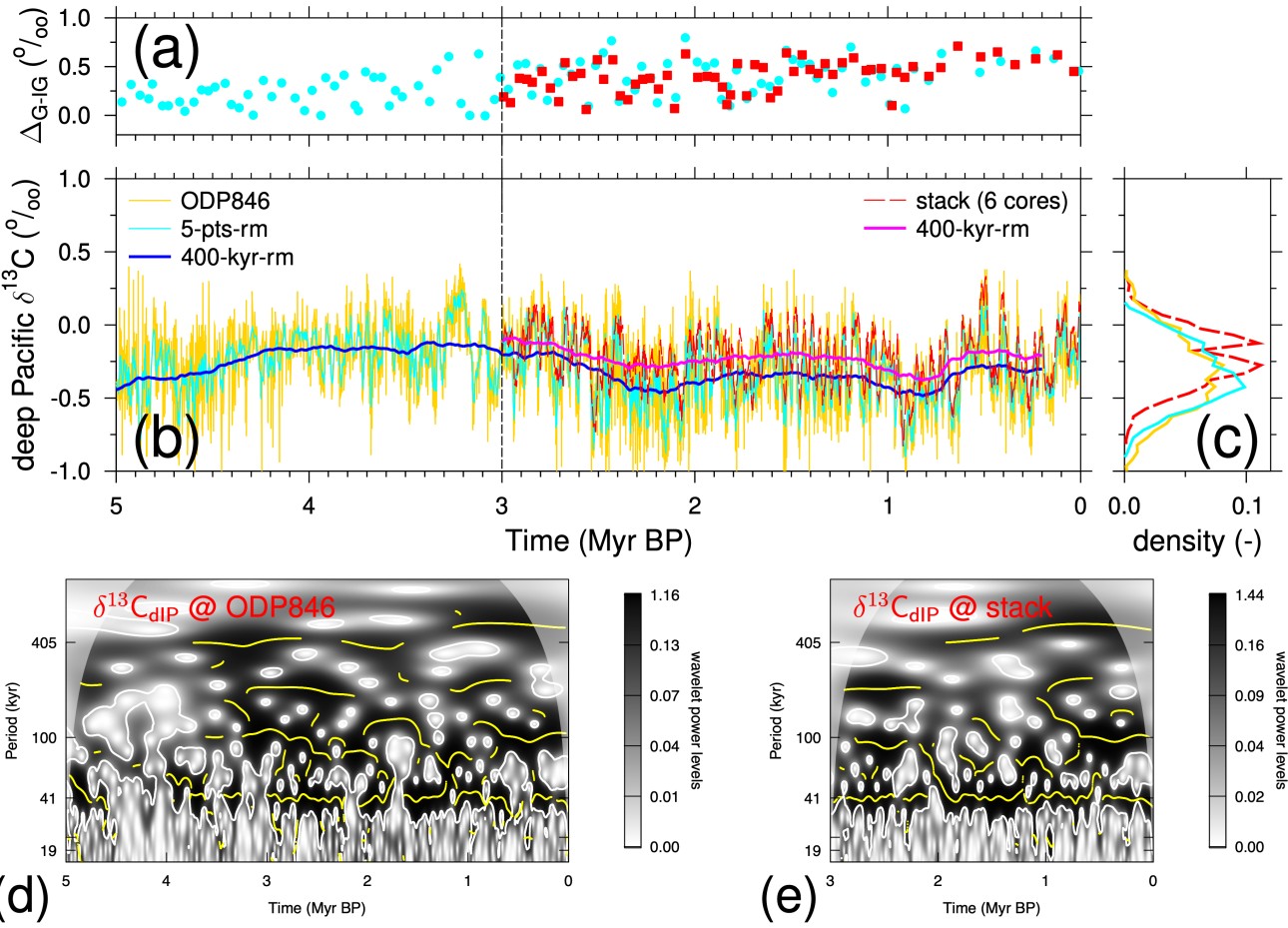

**Figure 1.** Deep Pacific Plio-Pleistocene $\delta^{13}$C data based on either ODP846 (Mix et al., 1995a; Shackleton et al., 1995; Poore et al., 2006), or a stack from six cores (Lisiecki, 2014). The temporal resolution is 1 kyr in the stack and partly up to 2 kyr in ODP846, which are interpolated to equidistant 1 kyr for further analysis. **(a)** G-IG amplitudes of records showing in (b). Here the difference between a glacial minima and the subsequent interglacial maxima are calculated following the MIS boundary definition of Lisiecki and Raymo (2005) with points being positioned at mid-transitions. **(b)** Time series. 400-kyr running means are added as is a 5-points running mean of ODP846 to improve comparison to the stack. The vertical line at 3 Myr BP marks the start point of the records for the analysis shown in (c). **(c)** Normalised data density distribution of the last 3 Myr of the $\delta^{13}$C time series shown in (b). **(d)** Wavelet of the 5-Myr long 5-points running mean of $\delta^{13}$C from ODP846. **(e)** Wavelet of the 3-Myr long deep Pacific $\delta^{13}$C stack.

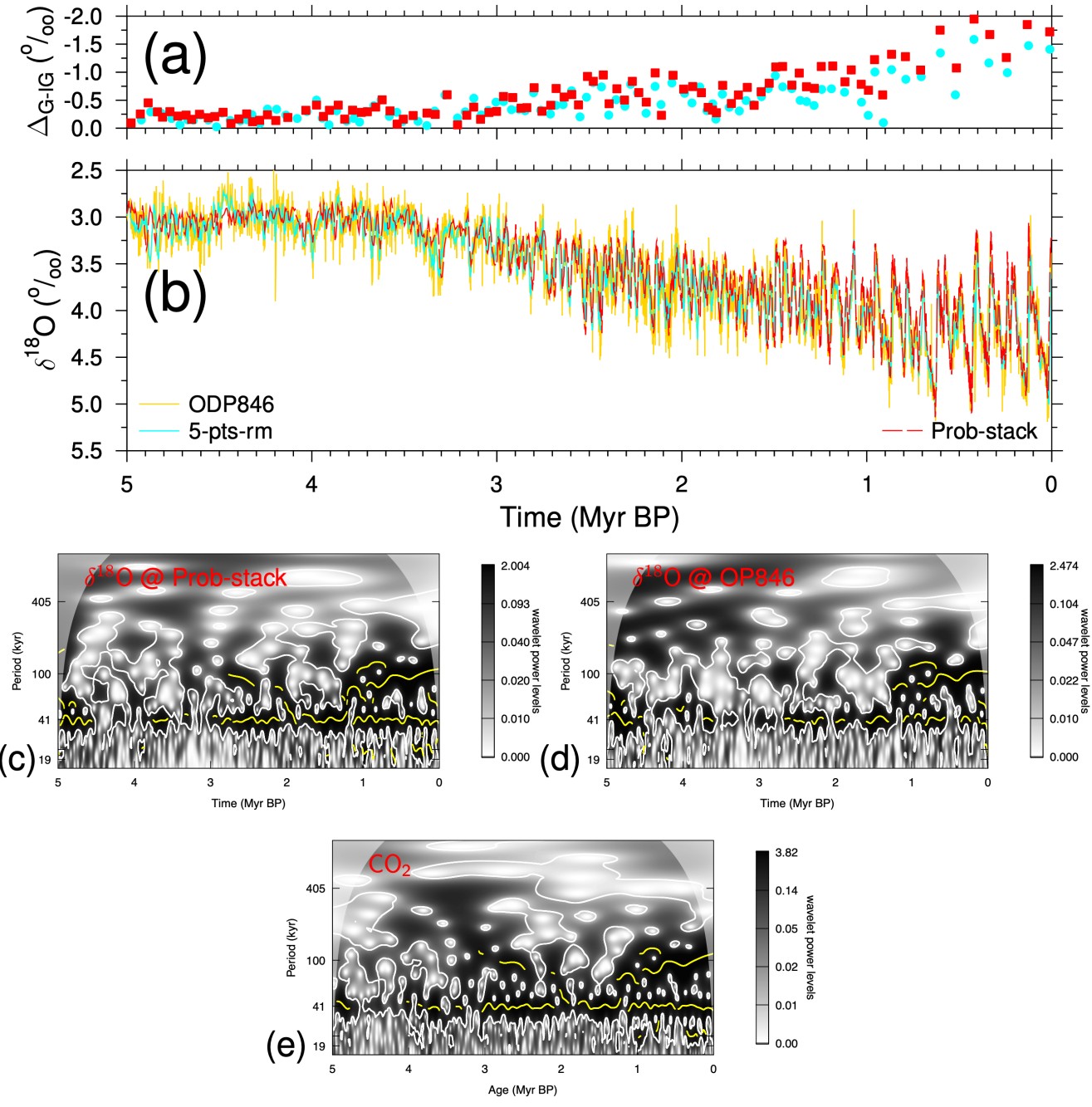

**Figure 2.** Climate change. (a) Glacial-interglacial amplitudes of records showing in (b). Here the difference between a glacial minima and the subsequent interglacial maxima are calculated following the MIS boundary definition of Lisiecki and Raymo (2005). Points are positions at the mid-transitions. (b) Benthic $\delta^{18}$O from ODP846 (Mix et al., 1995a; Shackleton et al., 1995; Poore et al., 2006) and Prob-stack (Ahn et al., 2017). Wavelets of detrended benthic $\delta^{18}$O (c): Prob-stack; (d): ODP846) and (e) simulated atmospheric $CO_2$ (scenario SEi++V6).

Spectral analyses of various climate variables find little of these slow changes during the Plio-Pleistocene. The Prob-stack (Ahn et al., 2017), the successor to the LR04 benthic $\delta^{18}O$ stack (Lisiecki and Raymo, 2005) based on 180 instead of 57 records, contains little power in 405-kyr periodicity during the last 2.5 Myr, but some power earlier-on (Fig. 2b,c), similarly as $\delta^{18}O$ in ODP846, the deep Pacific core from which $\delta^{13}C$ is used here (Mix et al., 1995a; Shackleton et al., 1995; Poore et al., 2006) (Fig. 2b,d). This long eccentricity cycle is also missing in the last 2.5-Myr and only weakly contained earlier in time series of my simulated atmospheric $CO_2$ (Fig. 2e). For the interpretation of the rather unusual frequencies around 200-kyr in $\delta^{13}C$ and the absence of the long eccentricity in $\delta^{18}O$ one should keep in mind that the response of a nonlinear system might exhibit frequencies not present in the original driven forcing (e.g. Rial, 1999; Rial et al., 2004).

Furthermore, during the Mid-Pleistocene Transition (MPT) around 1 Myr ago climate changed from a dominantly 41-kyr periodicity during the Pliocene and Early Pleistocene to an on average 100-kyr periodicity in the Late Pleistocene (e.g. Shackleton and Opdyke, 1976; Pisias and Moore-Jr., 1981). Since there is hardly any change in incoming solar radiation with power in the 100-kyr band (Laskar et al., 2004) this MPT is still not completely understood, but has so far been hypothesised to be caused by non-linear processes in the carbon cycle-climate system (e.g. Willeit et al., 2019; Berends et al., 2021; Clark et al., 2024). An accompanying feature of the MPT is the rise in G-IG amplitudes, as for example seen in benthic $\delta^{18}O$ (Lisiecki and Raymo, 2005; Ahn et al., 2017) or in global temperature compilations (Clark et al., 2024). Benthic $\delta^{13}C$ also includes this transition in power from 41-kyr to 100-kyr frequency across the MPT (e.g. Köhler and Bintanja, 2008). However, G-IG amplitudes in benthic $\delta^{13}C$ only gradually increase over the Plio-Pleistocene. The size of these amplitudes is in both the deep Pacific core ODP846 or the deep Pacific stack in the 41-kyr world on average 68–76% of their mean amplitude found in the 100-kyr world of the last 1 Myr (Fig. 1a). This is markly different from benthic $\delta^{18}O$ whose G-IG amplitudes in the 41-kyr world have been on average only 35-39% of their size during the last 1 Myr (Fig. 2a).

These two aspects — long eccentricity cycle and G-IG amplitudes — in which $\delta^{13}C$ differs from climate variables during the Plio-Pleistocene suggest that the carbon cycle and the climate system are on orbital timescales partly decoupled.

### 2.1.2 Late Pleistocene surface ocean $\delta^{13}C$ and atmospheric $\delta^{13}CO_2$

The continuous 155-kyr-long time series of atmospheric $\delta^{13}CO_2$ (Eggleston et al., 2016) has been shown to be highly correlated, especially on orbital time scales, to two mono-specific stacks of $\delta^{13}C$ from planktic foraminifera from latitudes $< 40°$, also called wider tropics (Köhler and Mulitza, 2024). This latter study could not identify any impact of the so-called carbonate ion effect proposed from laboratory experiments (Spero et al., 1997), which is why the planktic $\delta^{13}C$ stacks can be considered as reliable recorders of $\delta^{13}C$ in dissolved inorganic carbon ($\delta^{13}C_{DIC}$) in the surface ocean of the wider tropics. Underlying data were compiled from sediments cores at latitudes $<40°$, roughly in agreement with the non-polar (sometimes called equatorial) surface ocean boxes in the model. To my knowledge there exists no robust longer time series of planktic $\delta^{13}C$. Due to the shortness of these time series a spectral analysis with focus on the long eccentricity cycle is not possible. However, these data might help nevertheless via a comparison with simulation results. Recently, another $\sim$10-kyr-long part of $\delta^{13}CO_2$ across Termination IV has been published (Krauss et al., 2025).

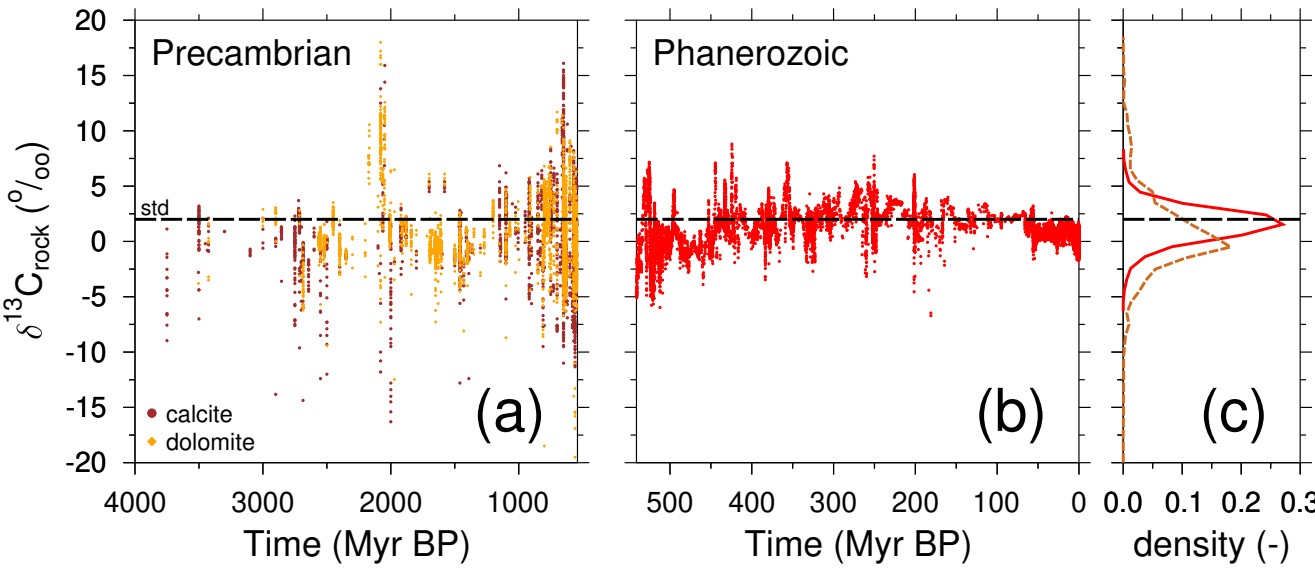

**Figure 3.** The $\delta^{13}$C from carbonate rock from Earth's history. (a) Precambrian values of $\delta^{13}$C calcite and dolomite rock as compiled in Prokoph et al. (2008). (b) Phanerozoic values as compiled in Bachan et al. (2017). (c) Normalised density plot of $\delta^{13}$C data (broken line: Precambrian; solid line: Phanerozoic). The standard (std) parameter value of $\delta^{13}$C$_{\text{rock}}$ = 2.0‰ used in the BICYCLE-SE model is also marked.

### 2.1.3 The $\delta^{13}$C signature of the geological source of carbonate weathering

Present day mountains have been built during the Phanerozoic (the last 540 Myr), but there exist also various areas with Precambrian shields (older than 540 Myr), e.g. in North America (e.g. Whitmeyer and Karlstrom, 2007; Faccenna et al., 2021). It is thus worth considering how $\delta^{13}$C in carbonate rocks ($\delta^{13}$C$_{\text{rock}}$) changed over these time spans. The records of $\delta^{13}$C in carbonates during the Phanerozoic (Fig. 3b) varied for most of the last 440 Myr between 0 and +5‰ with short-term excursions up to +8‰ and –5‰ (Bachan et al., 2017). The earliest part of the Phanerozoic (440–540 Myr) covering the Cambrian and
Ordovician contained in generally lower values (–5‰ to 0‰) with peaks up to +7‰ (Bachan et al., 2017). A compilation of raw whole rock $\delta^{13}$C isotope data for limestone and dolomite for the Precambrian (Fig. 3a), contains a huge scatter from –10‰ to +10‰ with a few excursions up to ±20‰ (Prokoph et al., 2008).

   Calculated density distributions of these time series of $\delta^{13}$C$_{\text{rock}}$ (Fig. 3c) are nearly normal distributed for the Phanerozoic with its mode being slightly lower (mean$\pm\sigma$ = +1.5 $\pm$ 1.6‰) than the chosen standard parameter values of $\delta^{13}$C$_{\text{rock}}$ = +2‰
in the BICYCLE-SE model (see section 2.2.1). Here, data have been interpolated to 10-yr equidistances before analysis. Data from the Precambrian have much larger gaps than for the Phanerozoic, which is why interpolation have not been applied. The distribution of the Precambrian $\delta^{13}$C$_{\text{rock}}$, combining both calcite and dolomite, is more widely than for the Phanerozoic with a small mean value (+0.7 $\pm$ 3.6‰).

### 2.1.4 The $\delta^{13}$C signature of volcanic $CO_2$

In BICYCLE-SE a $\delta^{13}$C of $-5$‰ of volcanic $CO_2$ emissions ($\delta^{13}C_v$) has been assumed, agreeing with a typical end member value from mantle material (Deines, 2002). However, $\delta^{13}C_v$ data are only available from modern observations and their variability in the past is unknown. Mid-ocean ridge basalt (MORB) volcanism contributes about 15% to the global volcanic $CO_2$ outgassing rate at present day (Burton et al., 2013) and contains in $\delta^{13}C_v$ a range from about $-9$‰ to $-4$‰ (Sano and Williams, 1996). The collected $\delta^{13}C_v$ values from more than 70 arc volcanoes (Mason et al., 2017) are on average ($\pm 1\sigma$) $-4.4 \pm 2.6$‰. The $\delta^{13}C_v$ range in arc volcanoes would narrow significant to an uncertainty of $\pm 0.5$‰, if weighted by $CO_2$ fluxes. However, such a weighting is not well constrained and $CO_2$ fluxes of different arc systems might have varied in the past. Newer data on arc volcanism from Baja California (Batista Cruz et al., 2019; Barry et al., 2020), not yet contained in the review of Mason et al. (2017), show volcanic $CO_2$ being more depleted in $^{13}$C with $\delta^{13}C_v$ being as low as $-19$‰. Similarly as MORB the arc volcanism is also responsible for about 15% of the annual volcanic $CO_2$ outgassing (Burton et al., 2013; Mason et al., 2017; Fischer and Aiuppa, 2020). Furthermore, new data on a 2021 volcanic eruption on Iceland (Moussallam et al., 2024) found a $\delta^{13}C_v = 0.1 \pm 1.2$‰. This is within the range of Icelandic $\delta^{13}C_v \in [-18.8, +4.6]$‰ (Barry et al., 2014), but on the upper end of the Gaussian distribution given by the $\delta^{13}C_v = -3.0 \pm 2.0$‰ by which the Icelandic compilation of Barry et al. (2014) is condensed in Mason et al. (2017). Indications for temporal variations of $\delta^{13}C_v$ in individual volcanoes also exist. For example, Chiodini et al. (2011) detected for the Etna a positive trend in $\delta^{13}C_v$, rising from $-4$‰ to $-1$‰ within 3–4 decades of the recent past. All-together, the collcted data suggest large heterogeneity of the mantle source, making a temporal fast changing $\delta^{13}C_v$ (as suggested here) a plausible possibility. However, although $\delta^{13}C_v$ data from arc volcanism and MORB are reasonable well covered the related $CO_2$ fluxes add up to only $\sim 30\%$ of the global volcanic outgassing. Thus, a complete picture of the variety of $\delta^{13}C_v$ including other sources, such as hot spot volcanoes, tectonic degassing and volcanic lakes (Burton et al., 2013; Fischer and Aiuppa, 2020), is still missing.

## 2.2 Model

### 2.2.1 The carbon cycle box model BICYCLE-SE

The carbon cycle box model BICYCLE-SE has been fully described in Köhler and Munhoven (2020) with its $^{13}$C cycle recently being updated (Köhler and Mulitza, 2024). This model version has recently been used to show that simulated changes in the radiocarbon age of the ocean between the LGM and preindustrial times agree reasonable well with reconstructions, although on millennial timescales (ignored here) abrupt changes in ocean circulation related to the bipolar seesaw need to be considered to meet deep ocean $^{14}$C data (Köhler et al., 2024). Briefly, the BICYCLE core of the model consists of 10 ocean boxes, seven boxes for terrestrial carbon pools and a one-box atmosphere. Carbon and the carbon isotopes (and alkalinity, $O_2$ and $PO_4^{3-}$ in the ocean) are calculated state variables of the model with explicitly considered carbonate chemistry in the ocean that distributes DIC as function of temperature, salinity and pressure into its three chemical species ($CO_2$, $HCO_3^-$, $CO_3^{2-}$). In its update to a solid Earth (SE) version the model now contains a process-based sediment module that calculates, depending on carbonate chemistry, either the accumulation or the dissolution of $CaCO_3$ in the deep ocean and the shallow water loss of

CaCO$_3$ due to coral reefs growth, simplistically calculated as function chemistry and sea level change. A fixed fraction of 0.6% of $^{13}$C-depleted organic matter of the export production of organic carbon that reaches the deep ocean boxes is permanently buried in the sediment ($35 \cdot 10^{12}$ gC/yr at preindustrial times). Thus, organic carbon is no active part of the sediment module, since its implementation would ask for much more complexity (Munhoven, 2021; Ye et al., 2025), potentially unavailable in box models. The main goal of the sediment module is to mimic the carbonate compensation feedback (Broecker and Peng, 1987), which according to a model/data comparison of deep ocean CO$_3^{2-}$ data (Yu et al., 2013) is sufficiently obtained (Köhler and Munhoven, 2020). The simulated export of CaCO$_3$ and of organic carbon to the sediment is by about 50% higher during glacial times than during interglacials, roughly in agreement with reconstructions (Cartapanis et al., 2016, 2018). Furthermore, carbonate and silicate weathering fluxes (on average 12 Tmol/yr each) introduce, either constantly or as function of atmospheric CO$_2$, a flux of bicarbonate (changing the total amount of both carbon and alkalinity) into the surface ocean. Volcanic CO$_2$ outgassing on land as a time-delayed function of changing land ice volume, or from island and hot spot volcanoes as a time-delayed function of sea level change (both delayed by 4-kyr, see Kutterolf et al. (2013)) are other external carbon sources to the model (about 5–15 Tmol/yr). More details are found in the earlier papers describing the model.

Isotopic fractionation occurs mainly during gas exchange and biological production on land and in the ocean. The sizes of the corresponding fractionations in the $\varepsilon$-notation (in ‰) are summarised in Fig. S1. Furthermore, the isotopic signatures of the external inputs of carbon in the simulated system — of the weathered carbonate rock ($\delta^{13}$C$_{\text{rock}} = +2‰$) and of the volcanic CO$_2$ ($\delta^{13}$C$_v = -5‰$) — are necessary parameters of the model, which so far have been constantly prescribed by the mentioned values.

Climate change (ocean circulation, temperature, sea level and sea ice, aeolian iron input) influencing the marine carbon pumps and land carbon storage is prescribed externally from reconstructions. I here use different setups, either covering the 800-kyr long time window with ice core data (Köhler and Munhoven, 2020) or the last 5 Myr with most of the Plio-Pleistocene (Köhler, 2023). Forcing details were described in the relevant studies, but are summarised in the SI (Figs. S2–S3).

In the model geometry all of the Indo-Pacific Ocean below 1 km depth is combined in one box called deep Indo-Pacific. All the relevant deep ocean $\delta^{13}$C data to which I compare my simulation results come from the deep Pacific. I always refer to either of these two elements when talking about the deep (Indo-)Pacific.

### 2.2.2 Reducing the model-data offset in the $^{13}$C cycle

Various applications have shown that the model seems to be able to simulate G-IG changes in the carbon cycle in reasonable agreement with various paleo data (atmospheric CO$_2$, deep ocean CO$_3^{2-}$, surface ocean pH, deep ocean $^{14}$C) (e.g. Köhler and Munhoven, 2020; Köhler, 2023; Köhler et al., 2024) while the long-term changes in $\delta^{13}$C as found in the reconstructions are so far not contained in a satisfactory manner in the model output (e.g. Köhler and Bintanja, 2008; Köhler et al., 2010; Köhler and Mulitza, 2024). Instead of searching for solutions which satisfy equally well the data constraints given by the carbon cycle itself and its stable carbon isotope I here follow a different, inverse, approach. I take the main carbon cycle including simulated atmospheric CO$_2$ as given and analyse how changes in the isotopic signatures of the external input of carbon in the simulated system need to vary to align the simulated $^{13}$C cycle, especially on orbital time scales, with data. In doing so I come to one

possible realisation of how the carbon cycle might have changed in the past, which reasonably well agrees with reconstructions of atmospheric carbon records ($CO_2$ and $\delta^{13}CO_2$) and marine $\delta^{13}C$, but I cannot exclude that other solutions might exist. Note, that the long-term effect of changes in the weathering flux strength, not its isotopic signature, on the $^{13}C$ cycle has recently been investigated with step-changes in the Bern3D model (Jeltsch-Thömmes and Joos, 2023), finding that equilibrium in the $^{13}C$ cycle is only reached after a few hundred thousand years, while that of $CO_2$ (and of climate) is reached an order of magnitude faster.

For that effort I apply additional constraints to the model. I overwrite (or prescribe) one internally calculated variable of the $^{13}C$ cycle with reconstructions. This would in principle violate mass conservation of $^{13}C$, but not if all necessary changes can be explained by variations in $\delta^{13}C_{\mathrm{rock}}$ or $\delta^{13}C_{\mathrm{v}}$, the isotopic signatures of the geological sources. In a post-processing analysis I determine their necessary values from model-internal information as described in the following. Initial results already informed me that when prescribing $\delta^{13}CO_2$ the necessary anomalies in isotopic signatures of the external carbon sources need to be a lot larger than the reconstructed ranges. In an alternative approach I therefore only weakly nudge the simulated $\delta^{13}C$ variable to its reconstruction. This approach can be used to check on necessary changes in either $\delta^{13}C_{\mathrm{rock}}$ or $\delta^{13}C_{\mathrm{v}}$ to align simulations with reconstructions. Both constraints (prescribing or nudging) are applied to either atmospheric $\delta^{13}CO_2$ or to deep Indo-Pacific $\delta^{13}C$. Since my solution is obtained from model/data differences (and not as a hypothesis to be tested that has been put into the model in a transient forward simulation) it can be called an inverse approach. Such inverse approaches are typically not leading to unique solutions but offer new, otherwise not (or hardly) available insights.

In detail, this approach works as follow: For each year $t$ the data would ask for a change $\Delta(t)$ (in units of ‰ per year) in one variable of the model ($\delta^{13}C_{\mathrm{model}}(t)$), e.g.

$$\Delta(t) = (\delta^{13}C_{\mathrm{data}}(t) - \delta^{13}C_{\mathrm{model}}(t)). \tag{1}$$

The difference of this data-model offset $\Delta(t)$ to the internally calculated change in this variable $\frac{\delta}{\delta t}(\delta^{13}C_{\mathrm{model}}^{\mathrm{internal}}(t))$, weighted by the nudging strength $\eta \in [0,1]$, is the applied correction $\Delta_{\mathrm{cor}}(t)$:

$$\Delta_{\mathrm{cor}}(t) = \eta \cdot \left( \Delta(t) - \frac{\delta}{\delta t}(\delta^{13}C_{\mathrm{model}}^{\mathrm{internal}}(t)) \right), \tag{2}$$

that is added to the differential equation

$$\frac{\delta}{\delta t}(\delta^{13}C_{\mathrm{model}}^{\mathrm{final}}(t)) = \frac{\delta}{\delta t}(\delta^{13}C_{\mathrm{model}}^{\mathrm{internal}}(t)) + \Delta_{\mathrm{cor}}(t). \tag{3}$$

If $\eta = 1$ these equations simplify to

$$\frac{\delta}{\delta t}(\delta^{13}C_{\mathrm{model}}^{\mathrm{final}}(t)) = \Delta(t) \tag{4}$$

and the approach describes the prescription of $\delta^{13}C$ with data. In other words, prescription is the most extreme case of nudging.

In the post-processing it is calculated how $\Delta_{\mathrm{cor}}(t)$ can be caused by changes in the isotopic signature of the external sources to the simulated carbon cycle. Thus, the following hypothetical isotopic signatures for the external fluxes are calculated:

$$\delta^{13}C_{\mathrm{rock}}^{\mathrm{hypo}}(t) = \frac{C(t)}{0.5 \cdot f_{\mathrm{wCa}}(t)} \cdot \Delta_{\mathrm{cor}}(t) + \delta^{13}C_{\mathrm{rock}}^{\mathrm{std}} \tag{5}$$

and

$$\delta^{13}\mathrm{C}_\mathrm{v}^\mathrm{hypo}(t) = \frac{\mathrm{C}(t)}{f_\mathrm{v}(t)} \cdot \Delta_\mathrm{cor}(t) + \delta^{13}\mathrm{C}_\mathrm{v}^\mathrm{std} \qquad (6)$$

Here, $\mathrm{C}(t)$ is the amount of carbon (in units gC) at time $t$ in the atmosphere, when nudging to atmospheric $\delta^{13}\mathrm{CO}_2$, or
in the deep Indo-Pacific ocean box, when nudging to deep Indo-Pacific $\delta^{13}\mathrm{C}$ with $f_\mathrm{wCa}(t)$ and $f_\mathrm{v}(t)$ being the annual fluxes
(in units of gC per year) of carbonate weathering and volcanic outgassing, respectively. The factor 0.5 in the denominator in
Equation 5 acknowledges that only half of the carbon in carbonate weathering has its source in weathered rock (e.g. Hartmann
et al., 2009). In my 800-kyr long simulations $f_\mathrm{wCa}$ is constant at 12 Tmol/yr (0.144 PgC/yr), while it varies as function of
$\mathrm{CO}_2$ between 10–15 Tmol/yr (0.12–0.18 PgC/yr) during the multi-millions year long runs. The volcanic $\mathrm{CO}_2$ outgassing flux
($f_\mathrm{v}(t) \sim$5–16 Tmol/yr or 0.06–0.192 PgC/yr) is a by 4-kyr time-delayed function of sea level change (Kutterolf et al., 2013).

I determined the strength $\eta_\mathrm{DP} = 0.0001$ when nudging to deep Indo-Pacific $\delta^{13}\mathrm{C}$ by comparing the resulting width of the
distribution in $\delta^{13}\mathrm{C}_\mathrm{rock}^\mathrm{hypo}(t)$ to the reconstructions (Fig. 3c). A comparable strength of the nudging is inversely related to the
amount of carbon in the relevant box. From the about 30 times more carbon in the deep Indo-Pacific box than in the atmosphere,
it was inferred that $\eta_\mathrm{A} = \eta_\mathrm{DP} \cdot 30 = 0.003$. For reasons of simplicity and due to a lack of data the same nudging strength is
used when calculating $\delta^{13}\mathrm{C}_\mathrm{v}^\mathrm{hypo}(t)$.

In my setup all $\mathrm{CO}_2$ from volcanic outgassing is directly released into the atmosphere. It has been shown (Hasenclever et al.,
2017) that submarine and subaerial injections of volcanic $\mathrm{CO}_2$ lead to similar changes in the carbon cycle on multi-millennial
time scale making a distinction of volcanism in the two subgroups irrelevant here. Important for long-term $\mathrm{CO}_2$ is the amount
of added carbon to the atmosphere-ocean-biosphere system, not the location of the injection. This is also the reason why I
investigate (a) how changes in $\delta^{13}\mathrm{C}_\mathrm{rock}$, which as part of weathering (or of bicarbonate injection) enters the system in the
surface ocean, might have an influence on atmospheric $\delta^{13}\mathrm{CO}_2$, and (b) how both weathering and volcanism might change
deep ocean $\delta^{13}\mathrm{C}$. Furthermore, these insights also suggest, that for cases of very strong nudging, as found in the prescribing
scenarios, results can probably not be explained by isotopic signature changes of the external carbon fluxes.

As final check for my hypothesis that the long-term changes in $\delta^{13}\mathrm{C}$ might be caused by the isotopic signature of the
external sources to the carbon cycle I compared $\delta^{13}\mathrm{C}$ from nudging scenarios with simulations in which either $\delta^{13}\mathrm{C}_\mathrm{rock}^\mathrm{hypo}(t)$ or
$\delta^{13}\mathrm{C}_\mathrm{v}^\mathrm{hypo}(t)$ as derived from Equations 5–6 is externally prescribed (Fig. S4). Differences between both setup are generally
less than 0.02‰ in various variables of the $^{13}\mathrm{C}$ cycle for the last 800 kyr supporting my post-processing approach to reliably
calculate the necessary isotopic signatures of these two fluxes in order to reduce the model-data offset in the $^{13}\mathrm{C}$ cycle.

### 2.2.3 Scenarios

My standard run (SEi) is based on the 800-kyr long scenario SE in Köhler and Munhoven (2020) with updates, mainly in
the $^{13}\mathrm{C}$ cycle, as described in Köhler and Mulitza (2024). Prescribing, or nudging to, atmospheric $\delta^{13}\mathrm{CO}_2$ (Eggleston et al.,
2016) is performed in scenarios C1 or C2, respectively. Here, simulations are started at 210 kyr BP in the interglacial around
MIS7a-c and the value of $\delta^{13}\mathrm{CO}_2$ before 155 kyr is assumed to stay constant. The whole 800-kyr long runs are alternatively
prescribed with (D1), or nudged to (D2), the deep Pacific $\delta^{13}\mathrm{C}$ stack. For an even longer perspective I rely on the 5 Myr-long

**Table 1.** Summery of applied simulation scenarios. The column "control" marks with "yes" the control runs or names the related controls.

| Name* | Length | Control | Description |
|---|---|---|---|
| SEi | 800 kyr | yes | Control run SE from Köhler and Munhoven (2020) with updated $^{13}$C cycle (Köhler and Mulitza, 2024) |
| C1 | 210 kyr | SEi | Atmospheric $\delta^{13}CO_2$ are **prescribed** from data (Eggleston et al., 2016) |
| C2 | 210 kyr | SEi | Atmospheric $\delta^{13}CO_2$ are **nudged** to data (Eggleston et al., 2016) |
| D1 | 800 kyr | SEi | Deep Indo-Pacific ocean $\delta^{13}$C are **prescribed** from the 6-cores stack of Lisiecki (2014) |
| D2 | 800 kyr | SEi | Deep Indo-Pacific ocean $\delta^{13}$C are **nudged** to the 6-cores stack of Lisiecki (2014) |
| SEi++V6 | 5 Myr | yes | Run SE++V6 from Köhler (2023) with strong weathering and 6% decline in volcanic outgassing 4–1 Myr with updated $^{13}$C cycle (Köhler and Mulitza, 2024) |
| D1-L | 3 Myr | SEi++V6 | Deep Indo-Pacific ocean $\delta^{13}$C are **prescribed** from the 6-cores stack of Lisiecki (2014) |
| D2-L | 3 Myr | SEi++V6 | Deep Indo-Pacific ocean $\delta^{13}$C are **nudged** to the 6-cores stack of Lisiecki (2014) |
| D2-P | 5 Myr | SEi++V6 | Deep Indo-Pacific ocean $\delta^{13}$C are **nudged** to the 5-points running mean $\delta^{13}$C from ODP846 as presented in Poore et al. (2006) |

∗: Naming convention: In non-control scenarios the first letter indicates what kind of data are used (C: atmospheric $\delta^{13}CO_2$; D: deep Indo-Pacific $\delta^{13}$C), the number indicates the form of data usage (1: prescribing; 2: nudging). Names with extended endings indicate 3-5 Myr long runs based on different data sets (-L: Lisiecki (2014),-P: Poore et al. (2006)).

runs published in Köhler (2023). Here, I choose a scenario based on run SE++V6, in which weathering is strongly coupled to atmospheric $CO_2$ and volcanic outgassing is decreasing by 6% between 4 and 1 Myr BP leading to a gradual decline in atmospheric $CO_2$. Tests have shown that the findings I describe further below are also robust for other scenarios included in Köhler (2023) with different weathering strength and volcanic history leading to alternative atmospheric $CO_2$ time series. Even when forced with the new 4.5 Myr long compilation of changes in surface temperature (Clark et al., 2024), which contains some power in periodicities slower than 100-kyr, my conclusions stay the same. I distinguish between a control run (SEi++V6), 3-Myr long runs in which deep Indo-Pacific $\delta^{13}$C is either prescribed with (D1-L) or nudged to (D2-L) the $\delta^{13}$C stack from Lisiecki (2014), and a 5-Myr long run (D2-P) nudged to $\delta^{13}$C in ODP846 as presented in Poore et al. (2006). An overview on all scenarios is compiled in Table 1.

## 2.3 Analysis tools

I use R (R Core Team, 2023) for calculating wavelets with WaveletComp (Roesch and Schmidbauer, 2018) and for further frequency analysis including coherence the package seewave, version 2.2.3. In all wavelet figures the white lines mark the 0.1 significant level and the yellow lines the ridges of the power distribution. Detrending of data has been performed with Matlab (The MathWorks Inc., 2023).

## 3 Results

Atmospheric $CO_2$ as simulated in my scenarios is shown as control (Fig. 4, first row) against data (Bereiter et al., 2015). This is chosen here as an indicator of the dynamics of the carbon cycle whose in depth discussions are covered in previous papers (Köhler and Munhoven, 2020; Köhler, 2023).

The control simulations (without prescription or nudging to $\delta^{13}C$) contain during the late Pleistocene typical multi-millennial variations of the order of 0.5‰ in atmospheric $\delta^{13}CO_2$ (Fig. 4, second row), of 0.3‰ in $\delta^{13}C_{DIC}$ of the wider tropical surface ocean (Fig. 4, third row), and of G/IG variations of 0.5‰ in $\delta^{13}C_{DIC}$ of the deep Indo-Pacific (Figs. 4, fourth row). As shown before (Köhler and Mulitza, 2024) atmospheric $\delta^{13}CO_2$ and $\delta^{13}C_{DIC}$ of the wider tropical surface ocean are on orbital timescales of 41-kyr periodicities highly correlated. In scenarios C1 (prescribed $\delta^{13}CO_2$) and C2 (nudged to $\delta^{13}CO_2$) there is also a loose correlation between atmospheric $\delta^{13}CO_2$ and deep Indo-Pacific $\delta^{13}C$ on these timescales (Fig. S5) with a coherence that rose from below 0.2 in scenario SEi to 0.6 (C1) or 0.4 (C2). A tighter correlation between $\delta^{13}C$ in the deep ocean and the atmosphere is not expected since the marine carbon pumps introduce vertical gradients in $\delta^{13}C$ (e.g. Schmittner et al., 2013) with opposite effects on atmospheric $\delta^{13}CO_2$ (or surface ocean $\delta^{13}C$) and deep ocean $\delta^{13}C$ on G-IG timescales (Köhler et al., 2010). Therefore, there has to be a certain anti-correlation between atmospheric $\delta^{13}CO_2$ / surface ocean $\delta^{13}C$ on the one hand and deep Pacific $\delta^{13}C$ on the other hand. In other words, prescribing or nudging to atmospheric $\delta^{13}CO_2$ or deep Pacific $\delta^{13}C$ always also increases the agreement of the simulations to the reconstructions in the other variable, but a perfect fit in the other endmember of the $^{13}C$ cycle is not expected. In scenarios C2 and D2 the long-term evolution from PGM-to-LGM in both atmospheric $\delta^{13}CO_2$ and wider surface ocean $\delta^{13}C$ is covered in the simulations, but the local maxima in the data around 80 kyr BP is not contained, suggesting that not these solid Earth fluxes but internal processes in the atmosphere-ocean-biosphere subsystem are responsible for these anomalies, which occur on shorter timescales. Millennial-scale changes in atmospheric $\delta^{13}CO_2$ between 70 and 60 kyr BP have already been successfully explained by such internal processes, namely rapid changes in terrestrial carbon pools and/or Southern Ocean air–sea gas exchange (Menking et al., 2022), which were probably also active around 80 kyr BP.

The introduction of the long-term periodicities into my simulated $^{13}C$ cycle can be seen in the spectral analysis. Wavelets show — apart from the time window 4–3 Myr BP — no power in periodicities beyond 100-kyr in the deep Indo-Pacific $\delta^{13}C$ in the control simulations SEi and SEi++V6 (Fig. 5a,b), but simulations contain power comparable to the data around 400-kyr but also in 200-kyr periodicities when nudged to the reconstructions (Figs. 1c,d and 5c,d). I therefore conclude that on these slow timescales my nudged simulations are in reasonable agreement with the data.

The time series of $\delta^{13}C_{rock}^{hypo}$ or of $\delta^{13}C_{v}^{hypo}$ necessary to close the $^{13}C$ cycle in the long eccentricity timescale are for the nudged scenarios (C2, D2, D2-L, D2-P) found in the two lowermost rows of Fig. 4, while those of the prescribed scenarios (C1, D1, D1-L) are due to the larger data scatter plotted with different y-axes in Fig. S6, which otherwise repeats Fig. 4. The resulting density distributions of both $\delta^{13}C_{rock}^{hypo}$ and $\delta^{13}C_{v}^{hypo}$ together with the data constraints are compiled in Fig. 6. When prescribing one $\delta^{13}C$ record from data (scenarios C1, D1, D1-L), it is clearly seen that the necessary distributions of the parameter values are with $1\sigma > 14‰$ ($\delta^{13}C_{rock}^{hypo}$) and $1\sigma > 9‰$ ($\delta^{13}C_{v}^{hypo}$) much wider than what reconstructions are suggesting ($1\sigma < 4‰$

for $\delta^{13}\mathrm{C_{rock}}$ and $1\sigma < 3‰$ for $\delta^{13}\mathrm{C_v}$ from present day arc volcanoes). This finding indicates, that a prescription of $\delta^{13}\mathrm{C}$, during which the model would follow the reconstruction not only on orbital timescales, but also during shorter multi-millennia anomalies, cannot be explained by the isotopic signature of either carbonate rock or volcanic outgassing. In other words, the scenarios C1, D1 and D1-L are unrealistic with respect to an explanation of the 405-kyr periodicity in the $^{13}\mathrm{C}$ cycle and can be discarded here.

In the nudged simulations (scenarios C2, D2, D2-L, D2-P) the resulting $\delta^{13}\mathrm{C_{rock}^{hypo}}$ and $\delta^{13}\mathrm{C_v^{hypo}}$ vary on orbital (and sometimes faster) timescales over the whole parameter ranges provided by the reconstructions (Fig. 4). While these dynamics challenge our understanding which processes might have been responsible for them (see Discussions) the resulting statistics (density distributions) are not unreasonable (Fig. 6). In detail, the nudged 800-kyr long simulations well overlap with the Phanerozoic reconstructions of $\delta^{13}\mathrm{C}$ in carbonate rock (D2: 0.8±3.0‰), while in longer simulations I find even tighter distributions, but with lower mean values (D2-L: 0.2±2.6‰; D2-P: –0.4±3.0‰) more in agreement with the Precambrian $\delta^{13}\mathrm{C}$ rock data (Fig. 6a,b). Similarly, the mean in $\delta^{13}\mathrm{C_v^{hypo}}$ is lower in nudged multi-millions years simulations than in those covering only 800-kyr (D2: –5.1±2.0‰; D2-L: –6.2±1.7‰; D2-P: –6.6±2.0‰) and the distribution are with $1\sigma \leq 2‰$ sufficiently narrow and overlap with data of $\delta^{13}\mathrm{C}$ in volcanic $CO_2$ (Fig. 6c,d). Remember, that the nudging strength $\eta_{\mathrm{DP}}$ was determined from the width of the distribution of $\delta^{13}\mathrm{C_{rock}^{hypo}}$ in scenario D2 to be comparable to the data, but its mean value was never used as tuning target, it freely evolved from the simulations and the post-processing analysis. The shift towards lower mean values in $\delta^{13}\mathrm{C_{rock}^{hypo}}$ or $\delta^{13}\mathrm{C_v^{hypo}}$ for longer simulations is dominantly caused by $\delta^{13}\mathrm{C}$ in ODP846 (used for nudging in scenario D2-P) being on average lower than $\delta^{13}\mathrm{C}$ in the deep Pacific stack (used for nudging in scenarios D2 and D2-L).

When nudged to $\delta^{13}CO_2$ (scenario C2) I obtain a rather skewed distribution of $\delta^{13}\mathrm{C_{rock}^{hypo}}$ (2.7 ± 2.6‰) which is wider than the reconstruction from carbonate rock (Fig. 6a). The width of the solution is not improving if I revise the nudging strength $\eta_A$ within reasonable bounds. Thus, it seems that the shortness of the $\delta^{13}CO_2$ time series, which covers not even half of one long eccentricity cycle is responsible for this unsatisfactory result. The distribution for $\delta^{13}\mathrm{C_v^{hypo}}$ (−4.5 ± 1.7‰) in scenario C2 has a slightly larger mean value but is in its width similar to what I obtain in scenario D2 from longer runs and nudging to deep Pacific $\delta^{13}\mathrm{C}$ (−5.7 ± 1.9‰). However, the shape of the distribution for C2 is without a clear maximum well different from a bell-shaped Gaussian curve of a normal distribution (Fig. 6c). Starting this analysis initially from understanding $\delta^{13}CO_2$ the available data are not yet covering a sufficiently long period for a complete understanding of the 405-kyr cycle in them. although the recently published new data around 340 kyr BP (Krauss et al., 2025) fully support my findings so far, where $\delta^{13}CO_2$ ranges between –6.8 and –7.3‰, much lower than in scenario SEi, but in agreement with my scenario D2.

As side effect, my nudging approach also increases the G-IG amplitudes in the simulated $^{13}\mathrm{C}$ cycle in the pre-MPT times. In the data and in the prescribed scenario D1-L the G-IG amplitudes in deep Indo-Pacific $\delta^{13}\mathrm{C}$ have prior to the MPT only been reduced to 75% of their post-MPT size, while in the control run SEi++V6 they have been reduced to 24% (Figs. 1a, 4c). My nudging approach increases these pre-MPT G-IG amplitudes to 41% (D2-L) and 32% (D2-P) of their amplitudes during the last 1 Myr, thus partially explaining this feature of the 41-kyr world (Fig. 4c).

All-together I conclude, that indeed both the carbon isotopic signature of either carbonate rocks or of volcanic $CO_2$ outgassing might be the process which closes the $^{13}\mathrm{C}$ cycle on periodicities slower than 100-kyr in the Plio-Pleistocene. Further-

more, the PGM-to-LGM offset in atmospheric $\delta^{13}CO_2$ and the long eccentricity in benthic $\delta^{13}C$ are potentially caused by the same processes and are therefore related to each other.

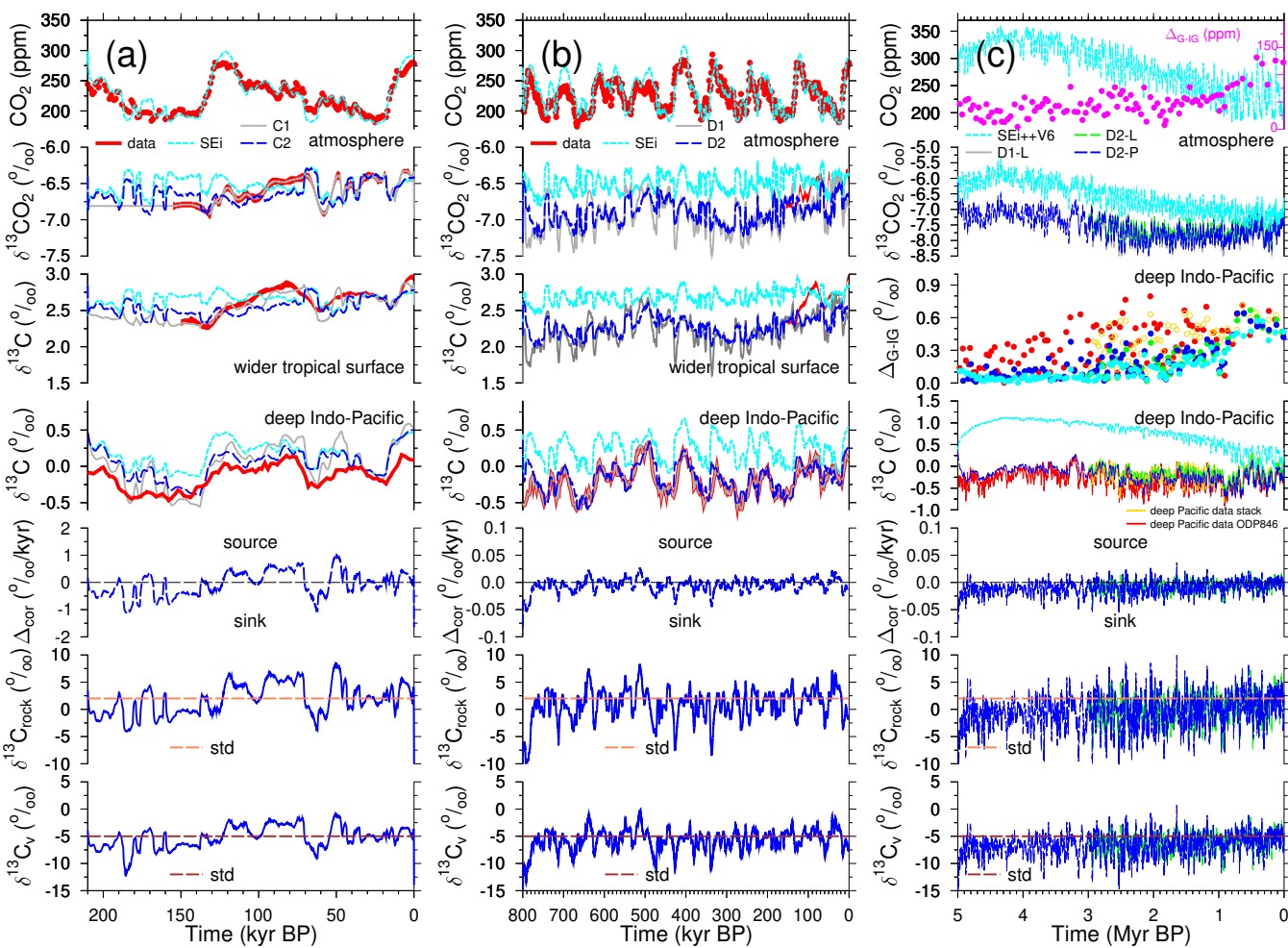

**Figure 4.** Closure of the $^{13}$C cycle on 405-kyr periodicity. From top-to-bottom: Atmospheric $CO_2$, $\delta^{13}CO_2$, $\delta^{13}C$ in DIC of surface water of the wider tropics, or in the deep Indo-Pacific, $\Delta_{cor}$, $\delta^{13}C_{rock}^{hypo}$, $\delta^{13}C_v^{hypo}$ for different scenarios including reconstructions of $CO_2$ (Bereiter et al., 2015), $\delta^{13}CO_2$ (Eggleston et al., 2016), $\delta^{13}C_{DIC}$ in the surface ocean of the wider tropics (anomalies to the LGM in the mono-specific stack of *G. ruber*, $\Delta(\delta^{13}C_{rub})$ from Köhler and Mulitza (2024)) or in the deep Pacific from the 6 cores stack (Lisiecki, 2014) or ODP846 (Poore et al., 2006). $\Delta_{cor}$ is calculated from (a) atmospheric $\delta^{13}CO_2$ or (b, c) from deep Indo-Pacific $\delta^{13}C$. In (c) $\Delta_{G-IG}$ are included for $CO_2$ (first row, right y-axis) and (third row) Indo-Pacific $\delta^{13}C$ (gold open circles for the 3 Myr-long deep Pacific $\delta^{13}C$ stack). No $\delta^{13}C$ in DIC of the wider tropical surface ocean is plotted in (c). For $\Delta_{G-IG}$ the difference between a glacial minima and the subsequent interglacial maxima are calculated following the MIS boundary definition of Lisiecki and Raymo (2005) with points being positioned at mid-transitions. The results in $\Delta_{cor}$, $\delta^{13}C_{rock}^{hypo}$ and $\delta^{13}C_v^{hypo}$ for the *prescribed* scenarios (C1, D1, D1-L) cover a wider range and are excluded here, but shown in Fig. S6.

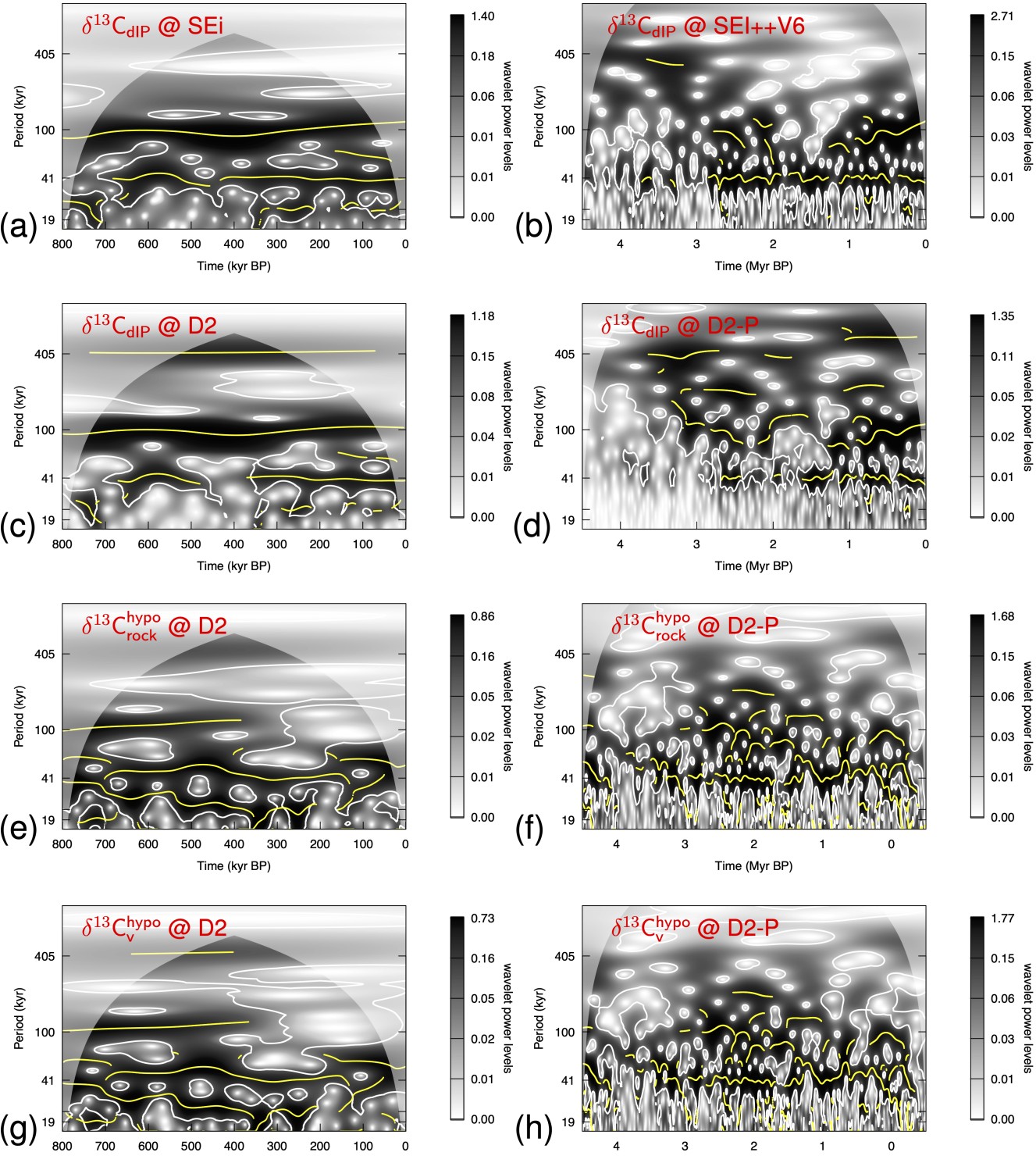

**Figure 5.** Wavelets of either the deep Indo-Pacific $\delta^{13}$C in (a–d) or of my solution for (e,f) $\delta^{13}C_{\text{rock}}^{\text{hypo}}$ and (g,h) $\delta^{13}C_{\text{v}}^{\text{hypo}}$ for different scenarios (left: SEi-D2 or D2; right: SEi++V6 or D2-P). Results are restricted to the last 4.5 Myr to avoid long spinup effects and only data in panel b have been detrended before spectral analysis.

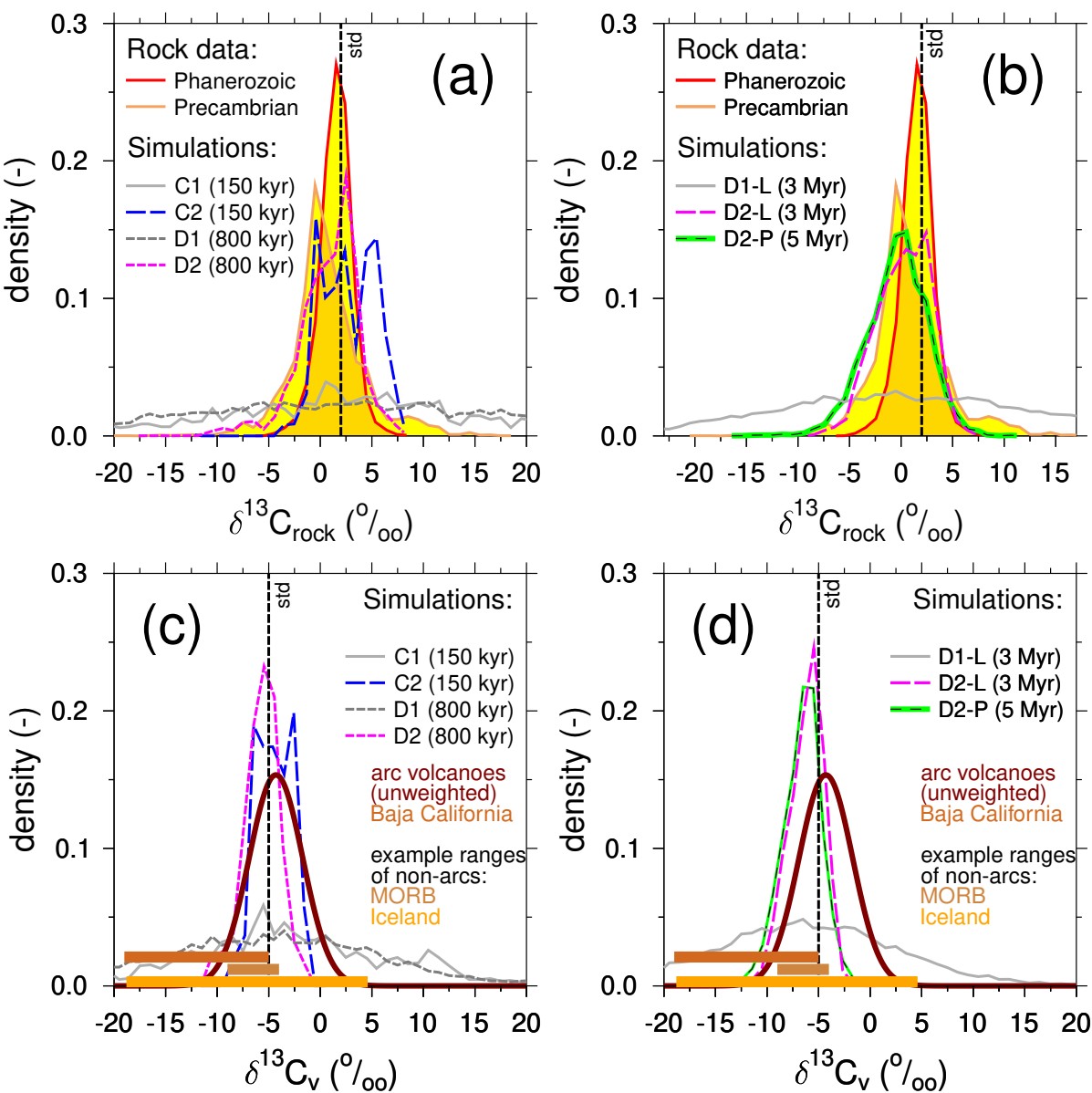

**Figure 6.** Distribution of carbon isotopic signature of (a, b) carbonate rock ($\delta^{13}C_{rock}$) and of (c, d) volcanic $CO_2$ ($\delta^{13}C_v$) from Earth's history in data and simulations. More details on the rock data are found in Fig. 3. Up to 800-kyr long simulations (a, c) versus multi-million years long simulations (b, d). The $\delta^{13}C_v$ data in (c, d) are from modern observations. The distribution of $\delta^{13}C_v$ data, not weighted by $CO_2$ fluxes, from the review of arc volcanoes (Mason et al., 2017) is shown here under the assumption of a normal distribution with mean$\pm 1\sigma$ of $-4.3 \pm 2.6$‰. Additionally, the range of data from newer studies on arc volcanism in Baja California (Batista Cruz et al., 2019; Barry et al., 2020) and the exemplary ranges from non-arc volcanism of mid-ocean ridge basalt (MORB) (Sano and Williams, 1996) and Iceland (Barry et al., 2014) are given. Vertical broken lines mark the standard (std) parameter values used in the model.

## 4 Discussions

A frequency analysis of the derived solutions for $\delta^{13}\mathrm{C}_{\mathrm{rock}}^{\mathrm{hypo}}$ and $\delta^{13}\mathrm{C}_{\mathrm{v}}^{\mathrm{hypo}}$ (Fig. 5e–h) finds that only little power is contained in periodicities beyond 100-kyr. Results for both variables are dominated by power in the obliquity band and some contributions to the 100-kyr band. Between 2–3 Myr BP some enhanced power in the 200-kyr periods is contained in both solutions obtained from scenario D2-P. Modifications in either $\delta^{13}\mathrm{C}_{\mathrm{rock}}^{\mathrm{hypo}}$ or $\delta^{13}\mathrm{C}_{\mathrm{v}}^{\mathrm{hypo}}$, both with little power around 405-kyr, lead nevertheless to a persisting occurrence of the long eccentricity in the resulting $^{13}$C cycle. A transition of power to other frequencies was suggested to be caused by the long residence time of carbon in the ocean during earlier quasi ice-free periods of the Cenozoic (Pälike et al., 2006; Zeebe et al., 2017). However, the power transition here had probably other reasons since the main carbon cycle and $CO_2$ contain no power in 405-kyr periodicity (Fig. 2g–i), which was not the case in these other studies. Looking closer at the spectra of the resulting simulated deep Indo-Pacific $\delta^{13}$C and the inversely obtained $\delta^{13}\mathrm{C}_{\mathrm{rock}}^{\mathrm{hypo}}$ (or $\delta^{13}\mathrm{C}_{\mathrm{v}}^{\mathrm{hypo}}$) for the same scenario I find, that the relative power in obliquity (41-kyr) and long eccentricity (405-kyr) is actually similar in both variables (Fig. S7). This similarity, not necessarily obtained from the related wavelets (Fig. 5d,f) supports the connection between both variables and highlights that obliquity is indeed most important in both time series, while 405-kyr is only weakly contained. Thus, it is not surprising that little 405-kyr power in deep Indo-Pacific $\delta^{13}$C is derived from $\delta^{13}\mathrm{C}_{\mathrm{rock}}^{\mathrm{hypo}}$ (or $\delta^{13}\mathrm{C}_{\mathrm{v}}^{\mathrm{hypo}}$) with similarly little relative power in the same frequency band.

To better understand how variations in $\delta^{13}\mathrm{C}_{\mathrm{rock}}$ or $\delta^{13}\mathrm{C}_{\mathrm{v}}$ with orbital frequencies impact on deep ocean $\delta^{13}$C I performed some tests. Using time series of the orbital parameters eccentricity, climate precession and obliquity (Laskar et al., 2004) artifical changes in $\delta^{13}\mathrm{C}_{\mathrm{rock}}$ or $\delta^{13}\mathrm{C}_{\mathrm{v}}$ were generated with patterns (mean, $\sigma$) similar to the reconstructions and then applied to some forward simulations modifying the control run SEi. The results (Fig. S8) show that $\delta^{13}\mathrm{C}_{\mathrm{v}}$ has a larger effect than $\delta^{13}\mathrm{C}_{\mathrm{rock}}$ on deep Indo-Pacific $\delta^{13}$C, probably because the related carbon influx ($CO_2$ outgassing) is about twice as large as that of carbonate rock during weathering. Furthermore, the resulting deep ocean $\delta^{13}$C anomalies with respect to SEi are lagging the forcing by about a quarter of the forcing periodicities. This is happening for all orbital forcings and frequencies indicating the time delay by which the deep ocean $\delta^{13}$C reacts to changes in the geological isotopic signatures. In other words, the change in $\delta^{13}\mathrm{C}_{\mathrm{rock}}$ or $\delta^{13}\mathrm{C}_{\mathrm{v}}$ obtained in the nudging approach, which is suggested to be responsible for the slow variations (405-kyr periodicity) in deep ocean $\delta^{13}$C is delayed accordingly. Although this lag might not be important for the spectra of $\delta^{13}\mathrm{C}_{\mathrm{rock}}$ or $\delta^{13}\mathrm{C}_{\mathrm{v}}$ it might need consideration if the hypothesis here is tested in future studies in forward simulations (see below). Furthermore, this lag suggests that the proposed solutions in $\delta^{13}\mathrm{C}_{\mathrm{rock}}$ or $\delta^{13}\mathrm{C}_{\mathrm{v}}$ might lead to some extent to over/undershooting in the target variable, which would influence the amplitudes but not the frequencies. However, since the obtained distributions in both variables are already in agreement with the reconstructions smaller ranges would only increase their match to the data.

The chosen setup here restricted my analysis to the inversely obtained changes in the isotopic signatures of the geological sources. Therefore, the changes in $\delta^{13}\mathrm{C}_{\mathrm{rock}}^{\mathrm{hypo}}$ and $\delta^{13}\mathrm{C}_{\mathrm{v}}^{\mathrm{hypo}}$ are not correlated to the underlying carbon fluxes ($r^2 < 0.01$), although volcanism was partly a function of the obliquity-dominated changes in sea level / land ice volume and the strength of the carbonate weathering in multi-million year long runs was related to $CO_2$, that is also dominated by obliquity (Fig. 2e).

Such relationships between fluxes and isotopic signatures, if found, might give further support to this hypothesis, since such

connections might be expected from a process-based perspective. However, previous model results have shown that changing the strength of fluxes without compensating effects very likely changes not only the $^{13}$C cycle but also the underlying main carbon cycle (Russon et al., 2010). Thus, such results are not easily obtained, but might be searched for in future studies.

Obliquity has an influence on incoming solar radiation in the high latitudes and is therefore thought to be the main control for land ice dynamics in the northern hemisphere (e.g. Huybers, 2007; Willeit et al., 2019; Köhler and van de Wal, 2020).

Thus, obliquity together with some influence of precession (Huybers, 2011; Barker et al., 2025) and internal feedbacks (e.g. Willeit et al., 2019; Berends et al., 2021; Clark et al., 2024) determines the G-IG cyclicity of the Plio-Pleistocene. Therefore, the dominance of obliquity in my solutions for either $\delta^{13}C_{\text{rock}}^{\text{hypo}}$ or $\delta^{13}C_{\text{v}}^{\text{hypo}}$ suggests that land ice might also play an important role in my closure of the $^{13}$C cycle for slow timescales of the Plio-Pleistocene.

Previously, the Pleistocene 400–500 kyr periodicity in marine $\delta^{13}$C has been suggested to be caused by continental weath-

395 ering and monsoon activity (Wang et al., 2010, 2014) and climate simulations also highlight the role of the 405-kyr periodicity on tropical precipitation (Yun et al., 2023). The Asia monsoon during the last 650 kyr as deduced from detrended $\delta^{18}$O in speleothems has significant power in obliquity, but in antiphase to summer insolation at 65°N, which suggests that the ultimate cause for the 41-kyr-related changes are situated in the northern hemispheric ice sheets (Cheng et al., 2016). Furthermore, present-day reconstructions of the continental weathering rates indicate that highly active regions dominate the global signal

with 10% of the area contributing about 50% of the global weathering fluxes (Hartmann et al., 2009). This study and others (Hartmann and Moosdorf, 2011; Moosdorf et al., 2011; Börker et al., 2020) also identified runoff, which is highly related to precipitation, as one of the dominant control of local weathering fluxes. Obliquity-controlled monsoon system have therefore a leverage on weathering, potentially activating contributions from different regions on Earth, with variable $\delta^{13}C_{\text{rock}}$, accordingly. I therefore suggest that obliquity-driven changes might also easily control weathering strength of different areas, finally

ending in changes in $\delta^{13}C_{\text{rock}}^{\text{hypo}}$ without the need for large variations in the global weathering flux itself. This influence of obliquity might operate via monsoon, but also a variable contribution of available weatherable rock around dynamic ice sheets in North America and Eurasia or from continental shelves during sea level low stands seems possible (e.g. Börker et al., 2020).

Dominant power in the obliquity band is also contained in the reconstructed activity of the circum-Pacific chain of arc volcanism, sometimes referred to as the "Ring of Fire", during the last 1 Myr (Kutterolf et al., 2013). In a revised analysis this

power in 41-kyr is reduced and only second behind 100-kyr (Kutterolf et al., 2019). This spectral analysis supports hypotheses that either changes in land ice load or in sea level might influence subaerial or submarine volcanism, respectively (Huybers and Langmuir, 2009, 2017; Hasenclever et al., 2017). Note, that in the model the component of volcanic $CO_2$ outgassing that is influenced by either land ice or sea level changes is lagging them by 4 kyr (Kutterolf et al., 2013). In the light of known temporal changes in the isotopic signature of individual volcanoes (e.g. Chiodini et al., 2011) and of the large heterogeneity of

the mantle material (e.g. Moussallam et al., 2024) the obliquity-related changes in $\delta^{13}C_{\text{v}}^{\text{hypo}}$ as inversely deduced here from the analysis are therefore also conceivable.

Possible ways of testing the proposed hypothesis of the obliquity-dominated influence on the isotopic signature of the geological sources might be the following. Most direct support might be derived from a forward modelling approach in which

the isotopic signature of the geological sources and the strength of the related fluxes are correlated to each other. This would, however, need a data-mining effort, in which fluxes and isotopic signature are spatially connected for present day. This approach might not generate long-time series as done here, but try to establish if fast changes in global mean $\delta^{13}C_{\mathrm{rock}}^{\mathrm{hypo}}$ or $\delta^{13}C_{\mathrm{v}}^{\mathrm{hypo}}$ as suggested here, are feasible from a more process-based understanding. However, unknown mantel heterogeneity and unknown temporal evolution of both local fluxes and local isotopic signatures might make this to a rather challenging task. Frequency analysis of various weathering proxies (e.g. Vance et al., 2009; von Blanckenburg et al., 2015) obtained from sediment cores in different ocean basins might give indications if on the regional scale obliquity played a dominant role, although this would give only indirect evidence on flux strength and not on $\delta^{13}C_{\mathrm{rock}}$. Monitoring programmes on present day volcanoes might give a broader foundation if temporal changes in $\delta^{13}C_{\mathrm{v}}$ as observed for the Etna are an exception or the rule (Chiodini et al., 2011). Finally, the connection of the PGM and LGM offset in atmospheric $\delta^{13}CO_2$ and how this is related to the 405-kyr periodicity might be verified with new ice core data of atmospheric $\delta^{13}CO_2$ further back in time.

## 5 Conclusions

Taken together my analysis, based on an inverse approach which nudged simulated $\delta^{13}C$ time series to reconstructions, suggests that variations in the isotopic signatures of the geological sources ($\delta^{13}C$ of weathered carbonate rock or $\delta^{13}C$ of volcanic outgassed $CO_2$) might without any further changes in the carbon cycle be sufficient to explain the signals in $\delta^{13}C$ related to the long eccentricity cycle during most of the Plio-Pleistocene. Furthermore, my suggested solution connects this periodicity around the 405-kyr in deep ocean $\delta^{13}C$ with the PGM-to-LGM offset in atmospheric $\delta^{13}CO_2$ implying that both are caused by the same underlying processes. My analysis highlights the influence of obliquity on these changes in the $^{13}C$ cycle, which I interpret as northern hemispheric ice sheets being the ultimate cause for the necessary variations in $\delta^{13}C_{\mathrm{rock}}$ or $\delta^{13}C_{\mathrm{v}}$. Studies show that both weathering and volcanic activity might, directly or indirectly, be influenced by land ice dynamics giving independent support for my hypothesis. Taken the link to obliquity as granted it is furthermore unlikely that the same mechanisms might have been responsible for the 405-kyr periodicity in marine $\delta^{13}C$ found in earlier climates of the Cenozoic since there has not been any substantial northern hemisphere ice sheet (Pälike et al., 2006; Zeebe et al., 2017; De Vleeschouwer et al., 2020; Westerhold et al., 2020). During these times not only $\delta^{13}C$, but also $\delta^{18}O$ and potentially atmospheric $CO_2$ contained cyclicity related to the long eccentricity which is missing here for the Plio-Pleistocene, also asking for other solutions to the problem. However, both my possible answers, either via carbonate weathering or via volcanism, seems similarly likely. So far, I have not yet found a criteria which makes one more likely than the other, and probably a mixture of both is the most realistic solution to the here suggested closure of the $^{13}C$ cycle on these slow timescales. However, as outlined in the discussion further investigations and proposed tests might favour one above the other. The dominance of obliquity in my solutions readily explains an improvement in simulated G-IG amplitudes in $\delta^{13}C$ in the 41-kyr world of the Plio-Pleistocene, but their reconstructed sizes are not fully obtained. Thus, while the influence of the isotopic signature of the geological sources might be important for the decoupling of climate from the carbon cycle there is room for other, not yet detected, processes for a complete understanding of the Plio-Pleistocene $^{13}C$ cycle.

*Data availability.* Simulation results data are available from PANGAEA (Köhler, 2025).

*Author contributions.* Since this is a one-author study all has been performed by the single author (PK).

*Competing interests.* The author declares no competing interests.

*Acknowledgements.* I thank Richard Zeebe for pointing the work of Hilgen et al. (2020) on the 200-kyr cycle to me, P.U. Clark for some comments on earlier versions of the draft and Florian Krauss and Jochen Schmitt for some insights on $\delta^{13}CO_2$ derived from ice cores. This publication contributed to Beyond EPICA, a project of the European Union's Horizon 2020 Research and Innovation Program (Oldest Ice Core). This is Beyond EPICA publication number 44.

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
