# Peer review of "Closing the Plio-Pleistocene 13C cycle in the 405-kyr periodicity by isotopic signatures of geological sources"

_Climate of the Past, 2024_

## Referee Comment (RC1)

*Review of Köhler, P., "Closing the Plio-Pleistocene $^{13}$C cycle in the 405-kyr periodicity by isotopic signatures of geological sources https://doi.org/10.5194/cp-2024-63 Preprint.*

**Summary**

Köhler presents a series of modelling experiments that quantify the possible impact of changes in the isotopic signature of either carbonate weathering or volcanic emissions on the carbon cycle over orbital timescales. The analysis forms the basis of a novel hypothesis that the mysterious presence of 400,000-year cycles in the $^{13}$C cycle are a product of changes in the isotopic composition of sources.

The paper will be an interesting result for the readers of Climate of the Past and particularly timely as it could inform new results from the soon to be extended ice core records.

[Figure]

The main result that isotopic composition of whole ocean-atmosphere-land biosphere system can change if you modify carbon isotope signature of the long-term carbon geologic fluxes (either volcanic or weathering) is not that surprising. What is surprising to me is the timescales over which these changes have the power to modify the whole ocean-atmosphere-biosphere system given the small fluxes (~0.1 PgC per year) into such a large system (~40,000 PgC) have enough leverage on the system (i.e. a ~400,000-year response time). See Figure 1 in this review where I spun a similar box model and then suddenly changed the isotopic composition of volcanic emissions. Yes, it changes the 13C of the whole system, but it takes a lot of time and input has to be large. Because of this weak lever, it means some of solutions presented in the analysis require one to really ratchet on the system by changing the isotopic composition of the source by a lot (sometimes up to 10 per mil in the study).

My primary criticism of the analysis is that I am left wondering how the isotopic composition of volcanic/weathering sources could shift (for long periods of time) by such wide ranges. I wasn't very convinced the mean isotopic composition could shift within the full range observed in either modern volcanic sources or the entire geologic record of carbonates. To shift the mean from one end of the distribution to the other, sometimes within a glacial cycle, seems extreme. Considering the volcanic hypothesis, it would suggest that at times in the past only small fraction of the modern-day volcanic regions are active, yet somehow as pumping out as much CO2 as today. Considering the weathering hypothesis, it would suggest that a relatively narrow geographic area might be contributing to the weathering flux. Overall, I would suggest some more work needs to be done to quantify if the swings in the source isotopic composition are feasible.

**Major comments:**

1. **Experimental design**

I struggled a bit to see the utility of the "prescribed/overwritten" experiments. I'm not sure they add much to the discussion. First, they are unphysical as they break conservation of mass and secondly, as the author acknowledges, are unrealistic as the swings in the isotopic composition are too large.  From an illustration point of view including them in the figures blows up the y axes (even when the axes are truncated which is not ideal). I would try and remove the prescribed experiments from the main text completely. This would also help the reader see the more realistic nudged experiments. In the current version I had to zoom in to maximum extent to see the important variability (my printed copy couldn't resolve the figures)

Secondly, it was quite tricky for me to pick out from the figures and the text exactly how big the impact the nudged experiments had on d13C budgets. It's kind of in the figures as the presumably the difference between the "control" runs the prescribed runs, but this is tricky to visualize.  Note one issues is that there is often a mean offset between the control and the nudged runs which I don't think is mentioned in the text. I would strongly suggest that somewhere the effect of the prescribed changes in isolation (i.e. just changing the external isotopic signatures) are shown.

2. **Why were changes in the mean isotopic composition of organic carbon not considered?**

By the logic that isotopic composition volcanic emissions or carbonate weathering is sufficiently wide to allow for major shifts back in time, it would follow that the changes in the weathering and/or burial of organic carbon, which also have a wide range of isotopic values could impact whole ocean-atmosphere-biosphere system.  From Cartapanis et al., 2018:  "We estimated the mean $\delta^{13}$C of organic matter in the first 10 cm of the sediment as −22.2 ‰ with a standard deviation of 2.3 ‰, consistent with prior literature (Sundquist and Visser, 2003)." Note however, that doesn't include the possibility of long-term changes in the C3-to-C4 abundance which seem possibly over these timescales)

Is there a reason that scenarios involving organic carbon were not considered? From the main body and supplementary information, it was not clear to me if long-term weathering and burial of organic carbon was included (there is a mention of the supplemental figure of phosphate burial). I tested this by running my model with the same experiments as in Figure 1 both with (shown) and without (not shown) a long-term weathering and burial flux of organic carbon (~0.06 PgC per year). Without the long-term fluxes I found the impact of changing the isotopic signature of volcanic sources was slightly greater additional gross fluxes.  Although a relatively minor effect, it would be good to know if they are included.

3. **Testing the plausibility of the scenarios**

One of the arguments for plausibility stems from the fact that the modelled changes in the source signature fall within the range observed distributions.

It would be more convincing if the changes in the isotopic signature of the weathering or volcanic fluxes were correlated with the modelled net fluxes.  Is this the case? For example, it would make more sense to me if you had a scenario where volcanic flux increased after deglaciation and also

changed the isotopic signature (perhaps reflecting more arc degassing and less mid-ocean ridge sources?).

On the weathering hypothesis, I can't think of any suggestions that are beyond the scope of this paper. I would suggest future work looking at the spatial pattern in the isotopic composition of riverine input to the ocean to see if obliquity-induced changes in the climate could cause major shifts in the delivery of $^{13}C$ to ocean.

**4. How are we going to test this hypothesis(es)?**

One of my major challenges with reading the paper was trying to figure out this this was a testable hypothesis. In some ways, it's a good challenge to have as it makes the reader think of new and interesting things to measure. However, I would have appreciated some more guidance. Moreover, it should be the case that when a new hypothesis is presented, we should set out ways to test it. Could the author please elaborate on the path forward in the conclusions?

**Line by line comments:**

Line Data 2.1. This section is too short and should either be folded in to below or more motivation for the compilations are needed. I suggest something to the effect of a list "To perform this analysis the following are needed:

Lines 77-98. Description of d18O and d13C decoupling conundrum. There's a lot of description the benthic d18O stack and its many features and thus a lot is packed in, including some very brief mentions of theories. Here I suggest a picture is worth a thousand words. Why not show the d18O stacks alongside the d13C (by combining Figure 1 with one the supplemental, possibly leaving the wavelet of d18O in the supplemental as it a better known).

Lines 125: Volcanic CO2 signature. *Mason et al., 2017* is a great up-to-date reference to use. Also, it may have ramifications for the analysis as the global emissions are estimated to range between at –3.8 to –4.6 per mil, slightly heavier than is typically assumed.

Mason et al., Remobilization of crustal carbon may dominate volcanic arc emissions. Science357,290-294(2017).DOI:10.1126/science.aan5049

Line 173. Please briefly note the conclusions of this study.

Line 178. "a lot bigger" = "larger/greater"

Table 1: The naming convention for the experiments is quite confusing as they don't contain any obvious information about whether they are nudged or prescribed. For examples, the letters -L and -P aren't clear to me. I would suggest a wholescale reset.

Line 290 "not a lot different" please rephrase with more precise language.

Line 312 ". Remember that in non-linear systems resulting frequencies might differ from those in the forcing (e.g. Rial et al., 2004)." I wasn't sure exactly what was being alluded to here. Can you elaborate more about the missing 400 ka cycle?

The premise of the analysis is that the 400ka cycle is difficult to explain and is missing from a modelled scenario, so I'm also a bit perplexed about why it does not then emerge from the analysis. Because you input (changes in rock or volcanic d13C) is passing through a massive damper, it would make sense that you would lose the higher frequency changes and those that are preserved would be heavily lagged. The fact that the results suggest the dominant power is around 40ka could suggest a number of things which I would appreciate some more discussion:

1) The control experiment (ie the SE) is missing crucial components in the 40 ka and thus the result is purely down to an inaccurate representation in the control.

2) The presence of the strong 40 ka cycle in prescribed/nudged experiments is a product of overfitting. If the model is trying to fit 400 ka signal with a very sluggish response time 100-400ka) there's a possibility that the model could overshoot and then undershoot if the tuning parameters are too sensitive. With the current analysis, I couldn't rule this out, but I believe the author could show with some simple experiments whether or not this is the case.

One possible way to help the reader understand what is going on would be to force the model is with synthetic timeseries of d13Cvolc/weathering with set periodicities and powers (e.g. an orbital curve with 400,100, 40, and 20 ka). Run that forcing through the model and see how the damper of the carbon cycle alters the resultant d13C in the atmosphere or deep Pacific. As example, here's a result using 65N summer insolation tied to large swings in the volcanic signature over the past 1 million year. This would also allow you to address how strong the lever is on the system. Alternatively, this could be done with the experiments I suggested above for isolating the effects of the prescribed isotope fluxes.

[Figure]

Line 390 "the ring of fire" please be more exact.

Figure 3. Colouring of data in panel C changes from red to grey and is confusing compared to other modelled scenarios also grey.

Sincerely,

Thomas Bauska

**References:**

Cartapanis, O., Galbraith, E. D., Bianchi, D., and Jaccard, S. L.: Carbon burial in deep-sea sediment and implications for oceanic inventories of carbon and alkalinity over the last glacial cycle, Clim. Past, 14, 1819–1850, https://doi.org/10.5194/cp-14-1819-2018, 2018.

Emily Mason et al., Remobilization of crustal carbon may dominate volcanic arc emissions.Science357,290-294(2017).DOI:10.1126/science.aan5049

---

## Author Comment (AC1)

**Response to Reviews**

**Referee #1:**

**Summary**

Köhler presents a series of modelling experiments that quantify the possible impact of changes in the isotopic signature of either carbonate weathering or volcanic emissions on the carbon cycle over orbital timescales. The analysis forms the basis of a novel hypothesis that the mysterious presence of 400,000-year cycles in the 13C cycle are a product of changes in the isotopic composition of sources.

The paper will be an interesting result for the readers of Climate of the Past and particularly timely as it could inform new results from the soon to be extended ice core records.

**My reply:** I thank Thomas Bauska for this very detailed review and his overall positive evaluation of the usefulness of this paper.

The main result that isotopic composition of whole ocean-atmosphere-land biosphere system can change if you modify carbon isotope signature of the long-term carbon geologic fluxes (either volcanic or weathering) is not that surprising. What is surprising to me is the timescales over which these changes have the power to modify the whole ocean-atmosphere- biosphere system given the small fluxes ($\sim$0.1 PgC per year) into such a large system ($\sim$40,000 PgC) have enough leverage on the system (i.e. a $\sim$400,000-year response time). See Figure 1 in this review where I spun a similar box model and then suddenly changed the isotopic composition of volcanic emissions. Yes, it changes the 13C of the whole system, but it takes a lot of time and input has to be large. Because of this weak lever, it means some of solutions presented in the analysis require one to really ratchet on the system by changing the isotopic composition of the source by a lot (sometimes up to 10 per mil in the study).

**My reply:** The scenarios with changes in the isotopic composition in the geological sources higher than 10 permil are scenarios, in which one time series of the model is identical to a data set (prescribed by a data set). The whole simulated $^{13}$C cycle then contains all changes — in all frequencies — in that prescribed time series and I discussed that this is rather unlikely and disqualify these prescribed scenarios as rather unrealistic. In the scenarios which are nudged to the data the changes in the isotopic signature of the geological sources are within the parameter ranges known from reconstructions (Figure 5) and therefore they are plausible solution. However, following some additional test suggested by the reviewer (see below) I now indicate that some over/undershooting of the inferred isotopic signature of the geological sources might indeed be part of the solution, which should not have an impact on the frequency distribution.

My primary criticism of the analysis is that I am left wondering how the isotopic composition of volcanic/weathering sources could shift (for long periods of time) by such wide ranges. I wasn't very convinced the mean isotopic composition could shift within the full range observed in either modern volcanic sources or the entire geologic record of carbonates. To shift the mean from one end of the distribution to the other, sometimes within a glacial cycle, seems extreme.

**My reply:** Single values of the isotopic signatures in the time series might give a false impression. I believe the **distribution** of the isotopic signature data (Figure 5) is a more powerful indicator. Extreme values in the nudged scenarios are rare and the obtained distributions are very similar to normal distributions, which I believe should be the case — and agree with the distributed reconstructions of the parameter values. Again, see below for the over/undershooting issue.

Considering the volcanic hypothesis, it would suggest that at times in the past only small fraction of the modern-day volcanic regions are active, yet somehow as pumping out as much CO2 as today.

**My reply:** We see from present day data, that the isotopic signature of volcanic $CO_2$, $\delta^{13}C_v$, might change drastically. For example, from the Etna it is known that $\delta^{13}C_v$ rose by 3‰ in 3-4 decades (Chiodini et al., 2011). This implies that there does not have to be a change in the activity of volcanoes emitting $CO_2$ to have a change in $\delta^{13}C_v$. Furthermore, the partial dependency of the volcanic $CO_2$ flux on changes in land ice or sea level as implemented in the model (see Figure F copying Figure 7 from Köhler and Munhoven (2020) below in responses to referee #2) already implies that changes in the source regions are assumed to have taken place. Additionally, new data on a 2021 volcanic eruption on Iceland (Moussallam et al., 2024) found a $\delta^{13}C_v = 0.1 \pm 1.2‰$. This is within the range of icelandic $\delta^{13}C_v \in [-18.8, +4.6]‰$ (Barry et al., 2014), but on the upper end of the Gaussian distribution given by the $\delta^{13}C_v = -3.0 \pm 2.0‰$ by which the icelandic compilation of Barry et al. (2014) is condensed in the review of Mason et al. (2017). This is another indication of large heterogeneities of the mantle source, making a temporal fast changing $\delta^{13}C_v$ (as suggested in the following) a plausible possibility.

Considering the weathering hypothesis, it would suggest that a relatively narrow geographic area might be contributing to the weathering flux.

**My reply:** This is indeed the case. For modern conditions it is know that highly active weathering regions in less than 10% of the land area contribute 50% of the weathering fluxes (Hartmann et al., 2009). This study furthermore showed that weathering rates are directly related to runoff, and therefore precipitation. Since weathering rates are highest in warm regions at low latitudes variable precipitation connected to shifting monsoon pattern give a direct control on how different areas (with different carbonate rock and different $\delta^{13}C_{rock}$) might get (de-)activated over time. Global weathering fluxes were assumed to be constant the simulated scenario SEi, but changed by about 20% in scenario SEi++V6.

Overall, I would suggest some more work needs to be done to quantify if the swings in the source isotopic composition are feasible.

**My reply:** Details along the lines mentioned above will be included in the revision, including the test how well-defined orbital-scale changes in the isotopic signature of the geological sources would impact deep ocean $\delta^{13}C$ and the possibility of over/undershooting.

**Major comments:**

**1. Experimental design**
I struggled a bit to see the utility of the "prescribed/overwritten" experiments. I'm not sure they add much to the discussion. First, they are unphysical as they break conservation of mass and secondly, as the author acknowledges, are unrealistic as the swings in the isotopic composition are too large. From an illustration point of view including them in the figures blows up the y axes (even when the axes are truncated which is not ideal). I would try and remove the prescribed experiments from the main text completely. This would also help the reader see the more realistic nudged experiments. In the current version I had to zoom in to maximum extent to see the important variability (my printed copy couldn't resolve the figures)
Secondly, it was quite tricky for me to pick out from the figures and the text exactly how big the impact the nudged experiments had on d13C budgets. It's kind of in the figures as the presumably the difference between the "control" runs the prescribed runs, but this is tricky to visualize. Note one issues is that there is often a mean offset between the control and the nudged runs which I don't think is mentioned in the text. I would strongly suggest that somewhere the effect of the prescribed changes in isolation (i.e. just changing the external isotopic signatures) are shown.

**My reply:** The prescribed experiments (C1, D1, D1-L) (first) do not break down mass conservation if all changes are coming from the isotopic signatures of the geological sources and (secondly) are necessary, because it is not clear from the beginning that they would lead to the large swings in the isotopic signatures as finally seen. Therefore, the comparison of the resulting $\delta^{13}C_{rock}$ or $\delta^{13}C_v$ distribution with the data (Figure 5) is one essential finding from which I can conclude that these prescribed experiments (C1, D1, D1-L) are unrealistic.

However, I see the problem of y-axes scale. A possible solution, which I like to implement in a revision, is, that in Figure 3, rows 5–7, the y-axes scales are changed for a better view on the nudged scenarios (C2, D2, D2-L, D2-P, blue lines). Results for the prescribed experiments (C1, D1, D1-L) will then be shown in a new figure with different y-axes scales in the SI.

The offset between the control and the nudged runs is consequently also seen in Figure 5 and discussed there when describing the mean and width of the different runs.

[Figure]

**Figure A:** Revised Figure 5 with changes y-axes in lower 3 lines.

**2. Why were changes in the mean isotopic composition of organic carbon not considered?**

By the logic that isotopic composition volcanic emissions or carbonate weathering is sufficiently wide to allow for major shifts back in time, it would follow that the changes in the weathering and/or burial of organic carbon, which also have a wide range of isotopic values could impact whole ocean-atmosphere-biosphere system. From Cartapanis et al., 2018: "We estimated the mean $\delta^{13}C$ of organic matter in the first 10 cm of the sediment as $-22.2‰$ with a standard deviation of 2.3 ‰, consistent with prior literature (Sundquist and Visser, 2003)." Note however, that doesn't include the possibility of long-term changes in the C3-to-C4 abundance which seem possibly over these timescales)

Is there a reason that scenarios involving organic carbon were not considered? From the main body and supplementary information, it was not clear to me if long-term weathering and burial of organic carbon was included (there is a mention of the supplemental figure of phosphate burial). I tested this by running my model with the same experiments as in Figure 1 both with (shown) and without (not shown) a long-term

weathering and burial flux of organic carbon ($\sim$0.06 PgC per year). Without the long-term fluxes I found the impact of changing the isotopic signature of volcanic sources was slightly greater additional gross fluxes. Although a relatively minor effect, it would be good to know if they are included.

**My reply:** Organic carbon is no active part of the sediments in BICYCLE-SE. However, I consider a fixed fraction 0.6% of the export production that reached the deep ocean to find its way to the sedimentary sink. This fraction was obtained in order to simulate a relatively constant overall $\delta^{13}$C in the simulated system. See also answers to referee #2 on this matter.
The sediment in the model accumulates $CaCO_3$ and is implemented as a simple, but still process-based module that can generate the carbonate compensation feedback, the slow dissolution of $CaCO_3$ if the carbonate chemistry in the deep ocean is asking for it.
The main intention of this paper here is to test the hypotheses that changes in the isotopic signature of the geological sources, as introduced in Schneider et al. (2013), might be responsible for the penultimate-to-last glacial maximum offset in atmospheric $\delta^{13}CO_2$ and how this might be related to the $^{13}$C cycle in general and the 405-kyr periodicity in particular.
The reason why organic carbon is not considered as active part of the sediment has also to do with the complexity which would then be neceesary in the sediment module (which would then be not anymore in balance with the overall rather simplistic model design). In detail, one wold need to calculate how dissolved chemical species (DIC, alkalinity, oxygen, etc.) enter and leave the sedimentary mixed layer by advection and diffusion and the reactions inbetween which would then decide on the fate of organic matter, if and how it might get remineralized by oxygen, and what happens in suboxic or anoxic conditions (Munhoven, 2021). I am involved in a project where such a complex sediment is coupled to a complex oceanic biogeochemical model embedded in an ocean GCM (Ye et al., 2023) and therefore decided here against a similar setup, also because the coarsely resolved ocean in my box model would not support the sediment module with the necessary details in the distribution of the chemical species.
Some of these details will be included in the revision.

**3. Testing the plausibility of the scenarios**
One of the arguments for plausibility stems from the fact that the modelled changes in the source signature fall within the range observed distributions. It would be more convincing if the changes in the isotopic signature of the weathering or volcanic fluxes were correlated with the modelled net fluxes. Is this the case? For example, it would make more sense to me if you had a scenario where volcanic flux increased after deglaciation and also changed the isotopic signature (perhaps reflecting more arc degassing and less mid-ocean ridge sources?).

**My reply:** Setups as suggested by the reviewer here are certainly possible next steps. However, I here refrain from them since I wanted to test first if the uncertainty in the parameter values of the geological sources are already sufficient to obtain the necessary changes. Furthermore, the strength of the present setup is that conclusions are based on the inversely obtained statistical output (which isotopic signatures does the model need and do the obtained results statistically agree with available reconstructions for the relevant parameter values?) and not on the process-based assumptions put into the model. In fact the suggested setup would be indeed an alternative way for testing if and how obliquity might influence weathering and/or volcanism. However, such setups are beyond the scope of the present paper. The previous paper of Russon et al. (2010) on the long eccentricity cycle in $\delta^{13}$C also suggested that such an approach will be difficult, since changing not only the isotopic signature but also the strength of carbon fluxes will always also impact the main carbon cycle and $CO_2$. Thus, my approach decouples the $^{13}$C cycle from climate / $CO_2$, which seems necessary, since the 405-kyr cycle is missing in the them. See also my reply at the end of the rebuttal of the comments from referee #1 on forward simulations versus inverse approaches.
Since the isotopic signatures and the geological fluxes have not been related during the setup, it is not

surprising, that they do not contain any correlation so far (see figures below for scenarios D2-L and D2-P).

[Figure]

**Figure B:** Calculated correlations between $\delta^{13}C_{rock}$ and the carbonate weathering flux and between $\delta^{13}C_v$ and the volcanic $CO_2$ outgassing flux for scenarios D2-L (top) and D2-P (bottom).

On the weathering hypothesis, I can't think of any suggestions that are beyond the scope of this paper. I would suggest future work looking at the spatial pattern in the isotopic composition of riverine input to the ocean to see if obliquity-induced changes in the climate could cause major shifts in the delivery of 13C to ocean.

**My reply:** Possible next step might be, as suggested by the reviewer (see previous point), implementing obliquity-related changes in weathering and its isotopic signature in the setup (forward modelling). Furthermore, different weathering proxies in different ocean basins might be analysed for obliquity. However, this is only an indirect check, it tests weathering strength of different areas but not changes in $\delta^{13}C$ of weathered rock.

**4. How are we going to test this hypothesis(es)?**

One of my major challenges with reading the paper was trying to figure out this this was a testable hypothesis. In some ways, it's a good challenge to have as it makes the reader think of new and interesting things to measure. However, I would have appreciated some more guidance. Moreover, it should be the case that when a new hypothesis is presented, we should set out ways to test it. Could the author please elaborate on the path forward in the conclusions?

**My reply:** Possible ways of testing the hypothesis would be
(a) forward modelling of obliquity-related changes in the isotopic signature of the geological sources and changes in the related fluxes and how results compare to data of the $^{13}C$ cycle;
(b) or, as mentionied above, frequency analysis of various weathering proxies in different areas. Although

this would give only indirect evidence and no direct idea how $\delta^{13}C_{\text{rock}}$ might have varied.

(c) For $\delta^{13}C_v$ it might help to better understand if the temporal change in the isotopic signature of the Etna volcanoe ($+3‰$ in 3-4 decades) is an exception or the rule. So, monitoring of $\delta^{13}C_v$ from individual volcanic $CO_2$ sources over decades might help to solve this issue.

(d) The connection of the penultimate-to-last glacial maximum offset in atmospheric $\delta^{13}CO_2$ and how this might be related to the $^{13}C$ cycle in general and the 405-kyr periodicity in particular can be tested with new ice core data of atmospheric $\delta^{13}CO_2$ and how they relate to my simulation scenarios. As mentioned already in the draft a 10 kyr long snapshot of $\delta^{13}CO_2$ around 350 kyr BP has now been measured and its publiciation is underway. I hope these new data, based on a recent PhD thesis of Krauss (2024) will be published before my paper here is finally accepted. If so, they will be included in my figure 3, since they give strong support for this connection.

The discussion will be expanded in this direction.

**Line by line comments:**

- Line Data 2.1. This section is too short and should either be folded in to below or more motivation for the compilations are needed. I suggest something to the effect of a list "To perform this analysis the following are needed:

  **My reply:** Following this comment the short introduction to subsection 2.1 will be expanded accordingly in the following direction: "To perform this analysis the following data (reconstruction) are needed: Plio-Pleistocene deep ocean time series of benthic $\delta^{13}C$ and $\delta^{18}O$, all available atmospheric $\delta^{13}CO_2$ from ice cores and late Pleistocene surface ocean (planktic) $\delta^{13}C$, and estimates on the variability in the $\delta^{13}C$ signature from carbonate rock weathering and volcanic $CO_2$ outgassing. These data are described in detail in this subsection."

- Lines 77-98. Description of d18O and d13C decoupling conundrum. There's a lot of description the benthic d18O stack and its many features and thus a lot is packed in, including some very brief mentions of theories. Here I suggest a picture is worth a thousand words. Why not show the d18O stacks alongside the d13C (by combining Figure 1 with one the supplemental, possibly leaving the wavelet of d18O in the supplemental as it a better known).

  **My reply:** I agree that the content of the supplement Figure S1 (climate change, mainly as seen in benthic $\delta^{18}O$) might be better placed in the main text. In the revision I will shift it accordingly. However, I do not think that both figures (1 and S1) should be combined to one single figure for various reasons. First, I like to keep the related wavelet analysis of $\delta^{18}O$ together with the plotted time series. Second, putting all together will only fit on one printed page if the sizes of the individual panels are reduced, which I like to avoid.

- Lines 125: Volcanic CO2 signature. Mason et al., 2017 is a great up-to-date reference to use. Also, it may have ramifications for the analysis as the global emissions are estimated to range between at −3.8 to −4.6 per mil, slightly heavier than is typically assumed. Mason et al., Remobilization of crustal carbon may dominate volcanic arc emissions. Science 357,290-294 (2017). DOI:10.1126/science.aan5049.

  **My reply:** Mason et al. (2017) is indeed a great reference for an overview of the range of $\delta^{13}C_v$ of $CO_2$ from arc volcanism (based on data from $>70$ individual arc volcanoes), which skipped my literature survey and which will be included in the revision. The related subsection 2.1.4, and the data constraints on $\delta^{13}C_v$ contained in Figure 5c,d will be revised accordingly, see new figures 5 c,d below.

[Figure]

**Figure C:** Revised distribution of $\delta^{13}C_v$. The $\delta^{13}C_v$ data are from modern oberservations. The distribution of $\delta^{13}C_v$ data, not weighted by $CO_2$ fluxes, from the review of arc volcanoes (Mason et al., 2017) is shown here under the assumption of a normal distribution with mean$\pm 1\sigma$ of $-4.3 \pm 2.6‰$.

Data compiled in Mason et al. (2017) contain a range in $\delta^{13}C_v$ between $-11$ and $0‰$ with its unweighted mean being $-4.3 \pm 2.6‰$. This range roughly agrees with the data from the references I so far included in section 2.1.4 with the exception of Baja California showing values in $\delta^{13}C_v$ as low as $-19‰$. However, Baja California data have been published in papers in 2019 and 2020, so after the publication of Mason et al. (2017).

- Line 173. Please briefly note the conclusions of this study.

  **My reply:** The main conclusion of the Bern3D model study of Jeltsch-Thömmes and Joos (2023) is, that equilibrium in the $^{13}C$ cycle is only reached after a few hundred thousand years, while that of $CO_2$ (and of climate) is reached an order of magnitude faster. These details might be included in the revision.

- Line 178. "a lot bigger" = "larger/greater"

  **My reply:** Changed to "larger".

- Table 1: The naming convention for the experiments is quite confusing as they don't contain any obvious information about whether they are nudged or prescribed. For examples, the letters -L and -P aren't clear to me. I would suggest a wholescale reset.

  **My reply:** Scenarios SEi and C1 are in identical form already described and to some extend analysed in Köhler and Mulitza (2024). Therefore, I believe their labels should not be changed. Furthermore, there is a logic to the names: Cx are scenarios in which **atmospheric** $\delta^{13}CO_2$ is forced/nudged to data, Dx are scenarios in which **deep Indo-Pacific** $\delta^{13}C$ is forced/nudged to the data. The number 1 in the name indicates **forcing** to data, the number 2 indicates **nudging** to the data. For the long runs (3-5 Myr) the ending "-L" indicates that the runs are forced/nudged to the 3 Myr-long $\delta^{13}C$ stack from **L**isiecki (2014), the ending "-P" indicates the runs are forced/nudged to 5 Myr-long $\delta^{13}C$ data from ODP8436 as presented in **P**oore et al. (2006). Thus, I will keep the names of the scenarios as is, but will add the logic of the naming convention to Table 1. Thinking about these names in detail I realized that the length of the scenarios D1-L and D2-L are given in Table 1 with 5 Myr, but they are only 3 Myr long. This will also be corrected. To ease the understanding of the scenarios for the reader I will also add the sources of the used data in Table 1.

- Line 290 "not a lot different" please rephrase with more precise language.

  **My reply:** This is will revised adding more details on the data plotted in the relevant figure to something like: "The distribution for $\delta^{13}C_v^{hypo}$ ($-4.5 \pm 1.7‰$) in scenario C2 has a slightly larger

mean value but is in its width similar to what I obtain in scenario D2 from longer runs and nudging to deep Pacific $\delta^{13}$C $(-5.7 \pm 1.9\text{‰})$. "

- Line 312 ". Remember that in non-linear systems resulting frequencies might differ from those in the forcing (e.g. Rial et al., 2004)." I wasn't sure exactly what was being alluded to here. Can you elaborate more about the missing 400 ka cycle?

**My reply:** Citing Rial et al. (2004) is indeed not well placed here. This sentence will be deleted. The 405-kyr cycle is missing in all my forcing files, so it is no surprise that it is also missing in the result files. Even in runs based on the new SST compilation of Clark et al. (2024), which contains some power in 405-kyr, the simulated $^{13}$C cycle is in the control runs not significantly moved towards 405-kyr (mentioned in section 2.2).

The premise of the analysis is that the 400ka cycle is difficult to explain and is missing from a modelled scenario, so I'm also a bit perplexed about why it does not then emerge from the analysis. Because you input (changes in rock or volcanic d13C) is passing through a massive damper, it would make sense that you would lose the higher frequency changes and those that are preserved would be heavily lagged. The fact that the results suggest the dominant power is around 40ka could suggest a number of things which I would appreciate some more discussion:

1) The control experiment (ie the SE) is missing crucial components in the 40 ka and thus the result is purely down to an inaccurate representation in the control.

**My reply:** As seen in the wavelets of the control runs (Figure 4a,b) this is not the case. The 41-kyr periodicity is dominantly contained in both control runs (SEi, SEi++V6).

2) The presence of the strong 40 ka cycle in prescribed/nudged experiments is a product of overfitting. If the model is trying to fit 400 ka signal with a very sluggish response time 100-400ka) there's a possibility that the model could overshoot and then undershoot if the tuning parameters are too sensitive. With the current analysis, I couldn't rule this out, but I believe the author could show with some simple experiments whether or not this is the case.

**My reply:** Over/underfitting is certainly a problem for the **prescribed** scenarios (C1, D1, D1-L) since the necesssary isotopic signatures of the geological sources were clearly out of the range provided by the data. However, for these prescribed scenarios I did not even perform wavelet analyses since the necessary parameter ranges gave enough indication that these simulations are unrealistic.
For the **nudged** scenarios (C2, D2, D2-L, D2-P) the calculated necessary values for $\delta^{13}$C$_{\text{rock}}$ or $\delta^{13}$C$_{\text{v}}$ were not varying extremely on suborbital timescale and their distribution meet the reconstructions. However, since I cannot produce more evidence on the plausibility of their changing rates I will — also based on the suggested test, see below — add some words on a potential contribution of over/undershooting to the draft, which might have an impact on amplitudes but not on frequencies of $\delta^{13}$C$_{\text{rock}}^{\text{hypo}}$ or $\delta^{13}$C$_{\text{v}}^{\text{hypo}}$.

One possible way to help the reader understand what is going on would be to force the model is with synthetic timeseries of d13Cvolc/weathering with set periodicities and powers (e.g. an orbital curve with 400,100, 40, and 20 ka). Run that forcing through the model and see how the damper of the carbon cycle alters the resultant d13C in the atmosphere or deep Pacific. As example, here's a result using 65N summer insolation tied to large swings in the volcanic signature over the past 1 million year. This would also allow you to address how strong the lever is on the system. Alternatively, this could be done with the experiments I suggested above for isolating the effects of the prescribed isotope fluxes.

**My reply:** Following this comment I performed some tests. Using eccentricity (E), E·sin($\omega$), and obliquity (which have the known orbital frequencies, see figure below) as baseline some artificial time series of $\delta^{13}$C$_{\text{rock}}$ or $\delta^{13}$C$_{\text{v}}$ were created, in which the means were replaced by the standard values of the model and the SD by the width of the reconstructions (Phanerozoic $\delta^{13}$C$_{\text{rock}}$ or $\delta^{13}$C$_{\text{v}}$ of arc

volcanoes, see Figure 5).

[Figure]

**Figure D:** Orbital parameters (a) and their spectral analysis (b) (Laskar et al., 2004).

If these artifical, orbitally-driven, time series of $\delta^{13}C_{rock}$ or $\delta^{13}C_v$ (right or left, respectively in figure below in thin lines) are used to force BICYCLE-SE the effect on deep Indo-Pacific $\delta^{13}C$ (thick lines) is seen in the difference to the control run SEi, in which these parameters stay constant on their standard values.

[Figure]

**Figure E:** Orbitally-forced isotopic signature of geological sources (thin) and responding deep Indo-Pacific $\delta^{13}C$ (thick), with difference (magenta) to the control run SEi.

What I take from these tests is the following: First, $\delta^{13}C_v$ has a larger effect than $\delta^{13}C_{rock}$, probably because the related carbon influx ($CO_2$ outgassing) is about twice as large as that of carbonate weathering. Second, the resulting deep ocean $\delta^{13}C$ anomalies with respect to SEi (magenta lines) are lagging the forcing by about a quarter of the forcing periodicities. This is happening for all orbital forcings and frequencies indicating the time delay by which the deep ocean $\delta^{13}C$ reacts to changes in the geological isotopic signatures. In other words, the change in $\delta^{13}C_{rock}$ or $\delta^{13}C_v$ obtained in the nudging approach, which is suggested to be responsible for the slow variations (405-kyr periodicity) in deep ocean $\delta^{13}C$ is delayed accordingly. Although this lag might not be important for the spectra of $\delta^{13}C_{rock}$ or $\delta^{13}C_v$ it might need consideration if the hypothesis here is tested in future studies in forward simulations. Furthermore, this lag suggests that the proposed solutions in $\delta^{13}C_{rock}$ or $\delta^{13}C_v$

might lead to some extent to over/undershooting in the target variable, which would influence the amplitudes but not the frequencies. However, since the obtained distributions in both variables are already in agreement with the reconstructions smaller ranges would only increase their match to the data.

My understanding of the dominance of the 41-kyr periodcity in $\delta^{13}C_{rock}$ or $\delta^{13}C_v$ is the following: As you can see the frequency analysis of the different orbital parameters (Figure D above) has for the climate precession ($E \cdot \sin(\omega)$) only power in periodicites related to precession ($\sim 20$ kyr), but none for those related to eccentricity (100, 405 kyr), although eccentricity (E) is part of the time series, which is clearly seen in Figure D(a). This simply expresses the relative importance of the different frequencies to the time series. (If the frequency analysis in Figure D(b) is plot on log scale one would see some diminishing powers on the order of $10^{-6}$ in the eccentricity bands.) It is actually a robust feature, not restricted to the applied method, similar results are obtained with different software and different spectral analysis methods. The same is happening here for $\delta^{13}C_{rock}$ or $\delta^{13}C_v$: the spectral analysis is dominated by the faster variations (41.kyr) and the much slower variations (405-kyr) have, in relative terms, little to contribute.

Another way to express this is plotted in the figure below, in which I compare the spectrum obtained over the whole 5 Myr from $\delta^{13}C_{rock}$ with that for deep Indo-Pacific $\delta^{13}C$. This is a condensation of the wavelets shown in Figure 5d,f, which contained quite some 405-kyr power in deep Indo-Pacific $\delta^{13}C$, but little in $\delta^{13}C_{rock}$. Here, both spectra are normalized to the power of the maximum peak (41-kyr), and show similar small contributions to the 405-kyr band.

So in summary, there is only little power in the 405-kyr band in the $^{13}C$ cycle, but for long-term effects this contribution neverthless needs to be accounted for. According to my paper here variations in the isotopic signatures of the geological sources, which have most power in the obliquity (41-kyr) band, are slowly shifting the system toward the 405-kyr band.

[Figure]

**Figure F:** Spectral analysis (normalized power) of $\delta^{13}C_{rock}$ and deep Indo-Pacific $\delta^{13}C$ of 5-Myr long time series from scenario D2-P.

Furthermore, one needs to keep in mind that solutions for $\delta^{13}C_{rock}$ or $\delta^{13}C_v$ were internally calculated after the described nudging procedure. This is an inverse approach, which is different to forward simulations. In the latter well defined time series with known variability drive the model which allows the analysis of leads and lags (as done here in the tests seen in Figure D), while the inverse answer to the problem is more statistically. Most of my former applications of the BICYCLE model to question of the carbon cycle of the past were following the forward approach, implying that the researcher has to have a hypothesis, which is tested by manipulating forcing data sets. This approach is only as good as the intuition of the researcher (combined with available research and computer time). The inverse approach is different in that sense, that it uses data and model to produce a suggestion (hypothesis) how boundary conditions might have changed to allow the model

to roughly follow the data. It gives new insights (as done here) since the upcoming suggestion would hardly have been produced with trial & error approaches necessary in forward simulations.
I will make clearer in the revision that I here follow an inverse apporach which might lead to new solutions, difficult to reach in conventional forward modelling.

- Line 390 "the ring of fire" please be more exact.

  **My reply:** Line 330 (not 390) contains the text passage about the "ring of fire". Assuming this is meant here, this text will be revised into "circum-Pacific volcanic chain, sometimes referred to as the "Ring of Fire"."

- Figure 3. Colouring of data in panel C changes from red to grey and is confusing compared to other modelled scenarios also grey.

  **My reply:** I will change the colours in Fig. 3c to be more in line with the other panels having data (reconstructions) in red. However, be aware that in the panel showing deep Indo-Pacific $\delta^{13}$C two data sets are shown in Fig. 3c (it was only one data set in the same panel in Figs. 3a,b). To distinguish both I have to use two colours (here: red+gold) for data.

Sincerely, Thomas Bauska

**Referee #2:**

Peter Köhler presents a series of box model experiments that explore the impact of changes in the isotopic signature of geological sources, related to carbonate weathering and/or volcanic emissions on the carbon cycle over orbital timescales. His model experiments reveal that the 400,000 periodicity observed in the 13C cycle (notably absent from any other climate variables) may have been related to changes in the isotopic composition of sources. Interestingly, the isotopic signatures for both sources required to best align the simulations with the existing benthic foraminifera 13C reconstructions, have most power in the obliquity band, suggesting that the ultimate forcing mechanism may relate to continental ice dynamics.

The manuscript is certainly a valuable contribution and should be published in Climate of the Past upon revision. It tackles an intriguing, yet outstanding, issue in paleoclimatology and offers a relatively simple solution. I am not a modeler myself, and I have to admit that the argumentation can be quite cumbersome to read in places. In particular, it would be great if §3 could be somewhat streamlined to make it easier to grasp for a wider audience.

**My reply:** I thank the referee for the overall positive evaluation of the draft. The section 3, which seemed to be difficult to grasp, is the result section, which describes the findings that are plotted in the main figures (3–5). It contains the core of the findings, which naturally is filled with details on which changes have been obtained. Changing this section might be difficult, however, due to the comments of both referees a much longer discussion section might now evolve in which results are set better into context.

**Major comments:**

I understand that you change the contribution of external (i.e. geological) sources of CO2 to the system (d13Crock and d13Cv) to understand how the global ocean mean d13C may have changed over time and whether these changes in the input term can reproduce the 405 kyr eccentricity cycles apparent in the Plio-Pleistocene marine d13C record. However, as I understand, the output terms (i.e. CaCO3 vs. Corg export and burial) are left to evolve untuned in the model? In particular, it is unclear to me whether the 13Corg composition is allowed to vary in space and time. Given that recent reconstructions showed that the burial of Corg and CaCO3 in pelagic sediments varied quite substantially (Cartapanis et al., 2016 & 2017), it is not clear to me how these output terms are considered in the box model.

**My reply:** Indeed, the export of $CaCO_3$ or organic carbon from the surface ocean boxes to the deep ocean boxes and further into the sediment is not changed in any of the analysed scenarios, they are all identical to their values in the control scenarios and have been discussed in detail in Köhler and Munhoven (2020). The $\delta^{13}$C signatures of all carbon fluxes, however, vary — i.e. is dynamically calculated internally — depending on the applied boundary conditions, here the change in the isotopic signature of the geological sources and how model results are prescribed or nudged to reconstructions.
Cartapanis et al. (2016) found higher glacial burial of organic carbon in sediments than during inter-glacials and Cartapanis et al. (2018) (I believe there is no article in 2017 from Cartapanis et al.) found higher glacial burial of $CaCO_3$ in sediments than during interglacials leading to changes in the total oceanic alkalinity budget. These dynamics are also found in the model and have been discussed in Köhler and Munhoven (2020), although without direct comparison to the papers of Cartapanis et al, see copy Figure 7 from that paper below for a scenario very similar to the control run SEi used here.

[Figure]

**Figure G (Fig. 7 in Köhler and Munhoven (2020)):** Contribution of solid Earth processes in standard run SE to changes in the global carbon cycle. The relevant process and the carbon species that is changes is mentioned in the legend. All fluxes apart from the volcanic outgassing lead to changes in both carbon in atmosphere-ocean-biosphere (AOB) and ocean alkalinity. Net ocean-to-sediment flux (yellow) and sediment accumulation (black) are plotted with a temporal resolution of 2 years (all other records with 100 years resolution) to visualize individual sediment dissolution events, that lead to a spiky net ocean-to-sediment $CaCO_3$ flux (difference between yellow points and black line highlighted by gray area). Volcanic outgassing and coral reef growth are plotted as 1-kyr running means.

Thus, these outputs to the sediment have not been motivated or forced by the Cartapanis et al. papers but are intrinsic to the model dynamics, which is why they might in details disagree with them, but the global dynamics of the glacial/interglacial changes are well matched. While there is a fixed ratio of $CaCO_3$:organic carbon in export and burial, the higher glacial export production directly also leads to higher fluxes to the sediment. The model output in Köhler and Munhoven (2020) agrees especially good in the difference between changes in $CO_3^{2-}$-ion concentrations in the deep Atlantic versus those in the deep Pacific as reconstructed in Yu et al. (2013) for the last glacial cycle being an indication that those deep ocean changes in alkalinity — and as consequences any carbonate compensation feedbacks (dissolution of $CaCO_3$ in upper sediments due to low $CO_3^{2-}$-ion concentrations) — are depicted reasonable well in the model.
See also reply to major comment #2 of referee #1 on organic carbon.
In the revision some of these details will be included in the draft to inform the reader on these dynamics.

Also, both d13Crock and d13Cv are characterized by abrupt (millennial-scale?), large-amplitude ($>$ 10 permil) oscillations across time (Fig. 3 bottom panels), which seem unrealistic to me. Can you maybe elaborate how these abrupt shifts can be explained? Are these related to high- vs. low-latitude erosion/weathering processes that vary on G-I timescales?

**My reply:** In this setup the isotopic signature of geological sources ($\delta^{13}C_{rock}$ or $\delta^{13}C_v$) are internally calculated by the the model in order to meet the dynamics of one of the $\delta^{13}C$ time series. When one of these $\delta^{13}C$ time series (either atmospheric $\delta^{13}CO_2$ or deep Pacific $\delta^{13}C$) is *prescribed* (scenarios C1, D1, D1-L) the necessary internally calculated values of $\delta^{13}C_{rock}$ or $\delta^{13}C_v$ need to vary abrupt and by up a lot (as mentioned by the referee by $> 10‰$, see grey lines in bottom panels in Figure 3). Due to these internally necessary large variations in the isotopic signatures which do not agree in their distributions with what we know from reconstructions, especially of $\delta^{13}C$ of carbonate rock (Figure 5) these *prescribed* scenarios have been discarded as unrealistic, implying that the fast changes in the $\delta^{13}C$ time series cannot be explained by changes in the isotopic signatures.
The alternative scenarios, in which the model is only *nudged* to the data (scenarios C2, D2, D2-L, D2-P),

as described in detail in the method section, leads to much smaller ($< 10‰$) and less abrupt variations in $\delta^{13}C_{rock}$ or $\delta^{13}C_v$ (blue lines in bottom panels in Figure 3), that agree in their distributions with what we know from reconstructions (Figure 5). These smaller and less abrupt changes in the isotopic signature can be easily be envisaged as possible (and plausible) changes in the source material of weathering or volcanism. Due to this agreement in the distribution with data these *nudged* scenarios are considered to be potentially possible model realisations of the $^{13}C$ cycle.

In the revision it will be made clearer that the *prescribed* scenarios which would imply large and abrupt changes in $\delta^{13}C_{rock}$ or $\delta^{13}C_v$ are unrealistic and no solution to the problem at hand.

**Minor comments:**

- p. 1, l. 13. Pre-Pliocene parts of the Cenozoic were largely ice free in the Northern Hemisphere only. Ice sheets have been present on Antarctica at least since the Oligocene.

  **My reply:** This sentence will be corrected accordingly.

- p. 5, l. 101 – not sure what is meant by "wider tropics"?

  **My reply:** In Köhler and Mulitza (2024) we compared data from low latitudes (about $< 40°$) with model output. This latitudinal range is roughly a combination of "tropics" and "subtropics", which is why we called it "wider tropics". I will add the meant latitudinal range to the term "wider tropics" in the revision.

- p. 7, l. 141. Maybe a naive question, but does the sediment model include Corg cycling or only CaCO3?

  **My reply:** As described in Köhler and Munhoven (2020): A fixed fraction of carbon of about 0.6% of the export production of organic carbon that reaches the deep ocean boxes is permanently buried in the sediment. This is for preindustrial climate $35 \cdot 10^{12}$ g C/yr. The exact amount has been determined by avoiding long-term trends in $\delta^{13}C$, since the loss of organic C (largely depleted in $^{13}C$) is counterbalancing the incoming, less $^{13}C$-depleted, geological fluxes. In the sediment module only $CaCO_3$ is followed and only sedimentary $CaCO_3$ can reenter the ocean due to its dissolution if wanted by the carbonate chemistry (carbonate compensation feedback). More details on the motivation and the content of the sediment module will be included in the revision.

- p. 11, l.263 – Can you briefly summarize what these internal processes are?

  **My reply:** This comment refers to Menking et al. (2022) and which processes *internal* to the atmosphere-ocean-biosphere subsystem have been found to explain millennial-scale changes in atmospheric $\delta^{13}CO_2$ between 70 and 60 kyr BP. The meant internal processes are "the superposition of rapid land carbon transfers and/or shifts in Southern Ocean air–sea gas exchange rates (perhaps modulated by sea ice)". This will be included in the revision.

- p. 16, l. 330-335. I would expect that changes in land ice load/sea level would induce a delayed response in subaerial/submarine volcanism due to the inertia of processes in the upper continual crust/mantle.

  **My reply:** Volcanism as implemented in the model (Köhler and Munhoven, 2020) has a constant component and a part that depends on changes in sea level / land ice. This part is indeed lagging the land ice / sea level changes by 4 kyr, as analysed in Kutterolf et al. (2013). This information was already contained in the draft, but it will be placed more prominently in the revision.

- Panels d and e in figure 1 as well as all panels in figure 4 are somewhat difficult to read. Wouldn't make sense to have these figures in colour?

**My reply:** The software used for this wavelet analysis (package WaveletComp in R) gives little room for changes in colour. There is actually, alternatively to b/w, one option in colour. However, in this alternative colourful plot the view is — due to the changes in colour — drawned to unimportant places. See example below replotting Fig 1d in both versions. In my view the colour version attracts to look especially where green switches to red, which in my view is not so important. Furthermore, I understood that this rainbow selection of colours is difficult to read for colourblind people and the guidances for authors in this journal ask the authors to be sensible to this problem when generating figures. Therefore, I decided to stick to b/w, since it offers a more objective visualisation of the pattern.

[Figure]

**Figure H:** Wavelet panel with b/w or rainbow colour-code.

**References**

[revised manuscript text omitted]